

# Internal solitary waves refraction and diffraction from interaction with eddies off the Amazon Shelf from SWOT

Chloé Goret[1], Ariane Koch-Larrouy[1], Fabius Kouogang[1,2], Carina Regina de Macedo [3,4],
Amine M'Hamdi[1,2,3], Jorge M. Magalhães[5], José Carlos Bastos da Silva[5,6], Michel Tchilibou[7],

Camila Artana[8], Isabelle Dadou [3], Antoine Delepoulle [7], Simon Barbot [3], Maxime Ballarotta [7],
Loren Carrère [7], Alex Costa da Silva[8]

[1] CECI, Université de Toulouse, CERFACS/CNRS/IRD, Toulouse, France

[2] Departamento de Oceanografia, Universidade Federal de Pernambuco, DOCEAN/UFPE, Recife, Brazil

[3] LEGOS, Université de Toulouse, CNES, CNRS, IRD, Toulouse, France

[4] Earth Observation and Geoinformatics Division, National Institute for Space Research (INPE), São José dos
Campos, Brazil.

[5] Interdisciplinary Centre of Marine and Environmental Research (CIIMAR), 4450-208 Matosinhos,Portugal.

[6] Department of Geoscience, Environment and Spatial Planning (DGAOT), Faculty of Sciences, University of
Porto, Rua do Campo Alegre, s/n, 4169-007 Porto, Portugal.

[7] Collecte Localisation Satellites (CLS), Ramonville-Saint-Agne, France

[8] LOCEAN-IPSL/CNRS, Université Pierre et Marie Curie,Paris, France.

*Correspondence to*: Chloé Goret (chloe.goret@cerfacs.fr)

**Abstract :** Off the Amazon shelf, mesoscale eddies interact with internal solitary waves (ISWs), modifying their
characteristics. For the first time, such interactions are directly observed through repeated measurements from a

set of high-resolution satellite data, including the recently launched SWOT (Surface Water and Ocean
Topography) mission. This study investigates ISWs detectable in SWOT Absolute Dynamic Topography (ADT)
and characterizes the changes in their properties induced by interactions with mesoscale eddies.

The analysis focuses on three scenarios: ISW propagation in the absence of eddies, ISW refraction by a cyclonic
eddy, and ISW diffraction by an anticyclonic eddy. ISW crests were identified and extracted using a band-pass

filtering technique, allowing accurate tracking of key features such as propagation direction, spacing between
individual crests, and wavecrest geometry. Before any interaction with eddies, mode-1 ISWs propagate steadily,
with consistent direction and planar wavefronts. A key finding is the variety of ISW responses depending on eddy
conditions. In the first case, in the absence of eddy, the interaction of ISWs with a seamount induced energy
transfer from mode-1 to mode-3 ISWs, while the propagation direction remains unchanged. In the second case, a

cyclonic eddy overlaying the seamount refracted ISW trajectories westward by approximately 50°, while also
increasing wavecrest curvature and enhancing the generation of mode-3 waves. In the third case, at the western
edge of an anticyclonic eddy near the seamount, the ISWs are split into two distinct paths: one branch refracted
westward, exhibiting flatter wave crests and reduced spacing between them; the other branch followed the eastern
edge of the eddy, displaying surface signatures of wave packets and enhanced wavecrest curvature.





These results demonstrate the effectiveness of the proposed approach in capturing the complex dynamics of ISWs. They offer novel insights into the nonlinear behavior of ISWs and their interactions with mesoscale and submesoscale oceanic features.



## 1 Introduction

Internal tides (ITs) are internal waves generated by the interaction of barotropic tidal currents with bathymetric features such as continental slopes, ridges, or seamounts, in stratified ocean. These baroclinic waves propagate in the ocean interior and can span hundreds of kilometers. When ITs propagate into regions of variable stratification or shallow topography, or when they encounter other waves or dynamical currents, nonlinear processes can cause them to steepen and disintegrate into trains of short, high-amplitude internal solitary waves,

ISWs (Jackson et al., 2012; Alford et al., 2015). ISWs often appear as wave packets and can propagate over long distances. The spacing between packets reflects the tidal forcing, ranging from over a hundred kilometers for mode-1 ITs to only a few kilometers for higher-order modes (De Macedo et al., 2023; Tchilibou et al., 2023; Le Mercier et al., 2012). Typically interfacial, ISWs travel horizontally along the seasonal or permanent pycnocline (Gerkema, 2001; Grisouard, 2011). The ISWs trajectories and properties are modulated by local environmental

factors—including background currents, mesoscale eddies, stratification, and bathymetric features—on timescales ranging from daily to interannual (Müller et al., 2012; Nash et al., 2012; Vlasenko et al., 2012; Magalhães et al., 2016; Liu and D'Sa, 2019; Tchilibou et al., 2022; Barbot et al., 2021).

ISWs are associated with strong vertical velocities and intense mixing, which impact the redistribution of physical and biogeochemical properties in the upper ocean (Assene et al., 2024; De Macedo et al., 2025; M'hamdi

et al., 2025). They contribute to energy cascades, air–sea exchanges, and ecosystem structuring (Sandstrom and Elliott, 1984; Huthnance, 1995; Munk and Wunsch, 1998; Muacho et al., 2013; Solano et al., 2023; Assene et al., 2024). ISWs also pose risks to offshore operations by destabilizing underwater structures and threatening navigation safety - a concern that is particularly relevant along the Brazilian Equatorial Margin, where a rapid expansion of oil and gas exploration is expected in the near future (Bole et al., 1994; Hyder et al., 2005;  He et al.,

2024). A better understanding of ISW–eddy interactions is thus essential for ocean energy budgets and hazard assessment, as eddies can transfer energy to higher modes and generate wave interference (Dunphy and Lamb, 2014; Ponte and Klein, 2015; Dunphy et al., 2017; Kouogang et al., 2025c in preparation).

The oceanic region facing the Amazon mouth constitutes a laboratory of experiment for studying IT and ISW interaction with dynamical mesoscale as the region is well known for IT and ISW generation and the

mesoscale activity induced by high dynamical currents (Fig. 1 and Fig. 2).

First, the region exhibits more than six IT generation sites along the Amazon Shelf break (Fig. 2, from A to F, Magalhães et al., 2016; Tchilibou et al., 2022), with the most energetics A and D that converge and B (Fig. 2). As they propagate, IT energy fluxes from A and D interact together and with the background environment, become unstable, and potentially disintegrate into ISWs packet that have been observed propagating several

hundred kilometers from the shelf break (Magalhães et al., 2016; de Macedo et al. 2023, Fig. 2). These waves probably cause intensified hot spots of mixing at more than 400 km from the shelf break (Kouogang et al. 2025a).

Second, off the Amazon, the region is influenced by the passage of an intense western boundary current, the North Brazil Current (NBC), which flows along the Brazilian coast (Fig 1).  This current forms a retroflection that feeds the North Equatorial Countercurrent  (NECC) and generates mesoscale activity with seasonal variability

(e.g. Aguedjou et al., 2019). Indeed, from March to July (MAMJJ), the pycnocline is shallow and the NBC is weak, while the river discharge of the Amazon River is high. Consequently, the internal tide flux remains relatively



stable and coherent. From August to December (ASOND), the pycnocline deepens, the river discharge decreases, and NBC intensifies (Silva et al., 2005; Aguedjou et al., 2019; Tchilibou et al., 2022), which forms NECC. Instabilities in these currents generate a series of cyclonic and anticyclonic eddies (Garzoli et al., 2004). These
structures significantly modify ISW propagation, trajectory, speed, amplitude, geometry and interpacket distance, and increase the incoherent part of IT (Bendinger et al, 2025 ; Dunphy and Lamb, 2014; Ponte and Klein, 2015; Dunphy et al., 2017; Wang and Legg, 2023).

Observational evidence of ISWs dynamics on sea surface height has long been limited by one-dimensional nadir altimetric measurements and by the low effective resolution of gridded multimissions altimetric
products, which are capable of detecting only oceanic features larger than several hundred of kilometers, especially close to the equator (Chelton et al., 2011; Ballarotta et al, 2019). The SWOT mission offers, for the first time, real-time, repeated and two-dimensional observations of the ocean surface of 2 km resolution with an effective resolution of approximately 7 - 10 km (low rate product) (Morrow et al, 2019).  SWOT can measure both sea surface height (SSH) with its Ka-band radar interferometer (KaRIn) and surface roughness (sigma0) using its SAR
radar. The combination of these datasets represents a significant advancement for the precise detection and tracking of ISWs (Fu et al., 2024; Morrow et al., 2019; Cheshmet al, 2025; Zhang et al., 2024). But this advance comes with new challenges. Currently, SWOT (Surface Water and Ocean Topography) data are corrected only for the stationary internal tide component using the High Resolution Empirical Tide (HRET) model (Zaron et al., 2019), and therefore still contain substantial nonstationary and ageostrophic signals—including ISWs. These waves partly
develop at wavelengths comparable to submesoscale structures such as fronts and filaments. A key challenge is to extract the ISW signal to study each process separately for modeling purposes and to accurately estimate geostrophic currents from SWOT measurements. Therefore, detecting ISWs and understanding their interaction mechanisms constitute both technical and scientific challenges due to their multiscale complexity and nonlinearity. In this study, we combine SWOT data, optical images acquired under sun glint conditions and daily MIOST
(Multiscale Inversion of Ocean Surface Topography) maps to explore the impact of eddies on ISW properties. Specifically, we examine changes in distance between crests, mode shifts, propagation directions, and wavecrest curvatures. Three cases are analyzed: (1) a reference case involving ISW propagation in the absence of eddies, (2) interaction with a cyclonic eddy and (3) interaction with an anticyclonic eddy. The paper is organized as follows. The satellites data are introduced in Sect 2. In Sect. 3, we present eddies and ISWs detection methods. Sect. 4
provides our résults. Next, Sect 5 presents discussion of the obtained results. Finally, a conclusion is provided in Sect. 6.

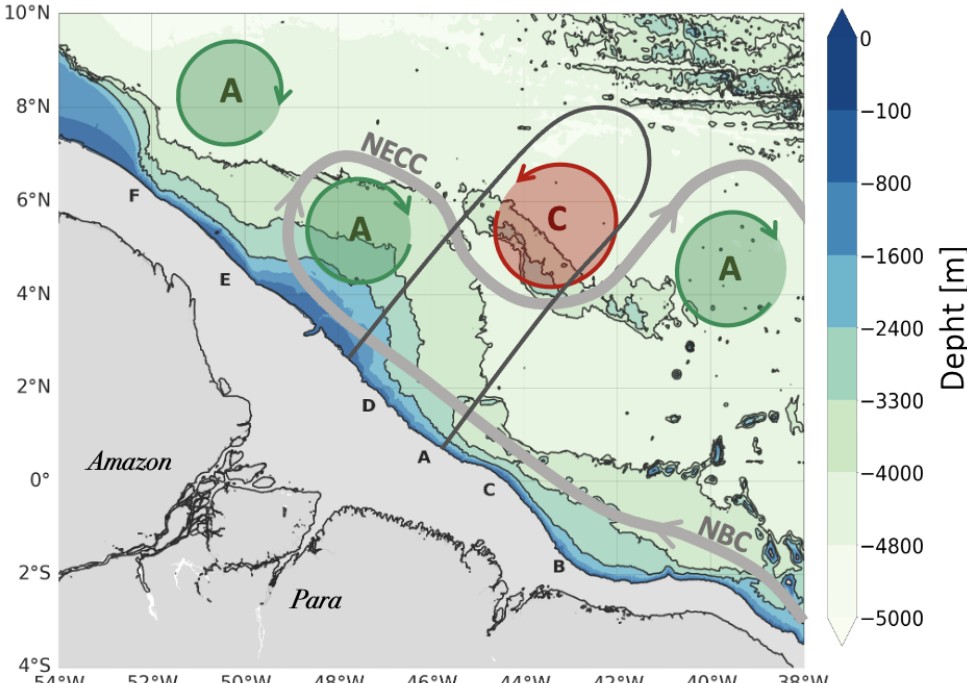

Figure 1 : Bathymetry of Amazon shelf from 0 to –5000 m. IT generation sites labeled A to F along the shelf break.
Black solid contours delineate a typical area where ISWs propagation is observed from sites A and D. The NBC and NECC are highlighted with thick grey arrows. Cyclonic eddies (CE) and anticyclonic eddies (AE) are marked respectively by red and green circles. Seamounts are delineated by 4000 m and 3300 m isobaths.



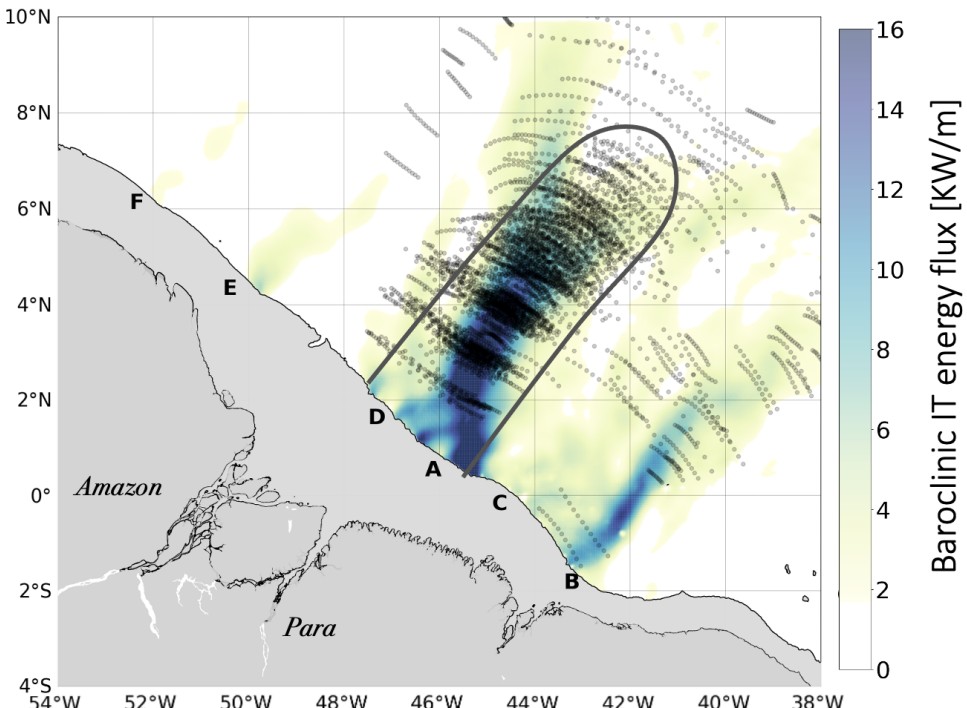

Figure 2 : The color map shows the 25-hour mean depth-integrated baroclinic internal tide energy flux from the
NEMO model from September 2015 (Assene et al., 2024), radiating from IT generation sites labeled A to F along
the shelf break. ISW surface signatures (black dotted lines) detected in MODIS/TERRA satellite imagery from De
Macedo et al. (2023). Black solid contours delineate a typical area where ISWs propagation is observed from sites
A and D.

## 2 Satellites data

Four complementary satellite datasets are used in the study, giving information on the location of real-time
mesoscale structures. These includes the absolute dynamic topography (ADT) and ocean surface roughness from
the new SWOT L3 KaRIn wide-swaths measurements, ADT maps from L4 multimission gridded product and
optical data acquired by MODIS TERRA/AQUA and NOAA-020 satellite. They are described in the following
sections.

### 2.1 L4 MIOST DT experimental

ADT maps are used to investigate the oceanic dynamics off the Amazon shelf and to detect mesoscale eddies. The
experimental daily MIOST maps are based on a multiscale, multivariate mapping of along-track altimetric
observations from several satellites, including SWOT (KaRIn and nadir), SARAL/AltiKa, CryoSat-2, HaiYang-
2B, Jason-3, Copernicus Sentinel-3A & 3B, and Sentinel-6A. These products are processed by SSALTO
(Multimission ground segment for altimetry orbitography and precise localization)/DUACS (Data Unification and



Altimeter Combination System) (Taburet et al, 2019) and distributed by AVISO with support from CNES (Centre national d'étude spatiales). The MIOST methodology (Ubelmann et al., 2021, 2022) enables improved reconstruction of ocean surface variability, particularly in delayed-time (DT) mode. This mode provides accurate

mapping of mesoscale structures and reduces mapping error by more than half compared to near-real-time (NRT) products. MIOST reaches a spatial resolution of 0.125° × 0.125° (Ubelmann et al., 2021b; Ballarotta et al., 2025). Off the Amazon shelf, the mapped wavelengths reach approximately 250–300 km, corresponding to processes with radius of about 70–90 km. The spatial resolution of the nadir altimeters used to generate these maps is too low to resolve submesoscale processes, such as ISWs.

**2.2 SWOT L3 product**

To overcome these constraints, wide-swath radar interferometry solutions were developed and deployed with the SWOT mission. The Ka-band Radar Interferometer (KaRIn), the central instrument of SWOT, is a Ka-band radar interferometer equipped with two SAR antennas positioned on either side of the satellite. This setup enables two-dimensional (2D) altimetric measurements across two lateral swaths, each approximately 50 km wide, providing

a total coverage of about 120 km along the track. KaRIn allows the resolution of ocean surface features at spatial scales around 15 km in wavelength (Morrow et al, 2019), about ten times finer than traditional gridded altimetry products. SWOT thus offers, for the first time, a snapshot of ISWs signatures in sea surface height (SSH), marking a significant advance in the study of their dynamics (Archer et al, 2025, Cheshm Siyahi et al, 2025). The orientation of the 21-day repeat cycle ascending passes is particularly well suited for observing tidal flows over the continental

slope of the Amazon. The swaths are nearly perpendicular to the coastline and align with the typical ISW propagation direction (Fig 1). We use the SWOT Level-3 SSH Expert product v2.0.1, derived from the Level-2 SWOT KaRIn low rate ocean data products (L2_LR_SSH). In order to obtain the total internal tide signal we sum the height of the sea surface anomaly unfiltered measured by KaRIn (ssha_karin_unfiltered), with all corrections and calibration applied and the coherent tidal correction from HRET (ssha_internal_tide) (Dibarboure et al., 2025).

Then, the absolute dynamic topography (ADT_swot) is reconstructed by adding the mean dynamic topography (mdt_karin) (Jousset et al, 2025). In order to highlight high-frequency signals containing ISWs signatures, the MIOST ADT interpolated to SWOT resolution (adt_miost) is subtracted. Therefore, ADT_swot contains all the high-resolution signal not resolved by the corrections applied to the KaRIn data, as well as the part of the signal not resolved by the MIOST mapping method.

ADT_swot = ssha_karin_unfiltered + ssha_internal_tide + mdt_karin - adt_miost

Another key measurement for observing ISWs is the surface roughness variation captured by the SAR radar backscatter. Joint analysis of SAR images (sigma0) and SWOT ADT enables the distinction between true soliton signals and other mesoscale or submesoscale structures. Thus, observing roughness contrasts facilitates the detection of soliton features.

**2.3 MODIS TERRA, MODIS AQUA and NOAA-020**

To complement the limited spatial coverage of the SWOT dataset, which is limited to the swath width (~120 km), we use one optical data for each case captured by different satellites. For no eddy case (NE) and cyclonic eddy



case (CE), we used images captured by the MODIS (Moderate Resolution Imaging Spectroradiometer) instruments onboard the Terra and Aqua satellites, respectively (doi: 10.5067/MODIS/MYD021KM.061). MODIS Level-1B
dataset are accessible through NASA's Earth Science Data System (ESDS) (De Macedo et al., 2023). The measurements are acquired on band 6, centered at 1640 nm, with a spatial resolution of 500 m. For the anticyclonic eddy case (AE), we utilized VIIRS Level 1-B calibrated radiance product (Visible Infrared Imaging Radiometer Suite) data acquired on bord the NOAA-20 satellite (doi: 10.5067/VIIRS/VJ102MOD.021), which captures images at 750 m spatial resolution. Unlike the MODIS Level-1B product, which covers a 5-minute time span, the VIIRS
Level-1B calibrated radiance product has a nominal temporal duration of 6 minutes. These datasets highlight variations in ocean surface roughness. Under sunglint conditions, where solar reflections enhance contrasts, optical images reveal the signatures of solitons at the water surface. However, the number of usable observations is significantly limited by cloud cover, as well as the location and angle of solar reflection (De Macedo et al., 2023).

## 3 Methods

### 3.1 Eddy Detection Method

Mesoscale eddies were detected from ADT fields derived from MIOST Level 4 products. To remove large-scale structures, we first applied a two-dimensional Lanczos filter to the ADT, with a cutoff wavelength of 1000 km in both latitude and longitude. This filtering highlighted mesoscale processes with a clear sea surface signature. Eddy detection was performed using the py-eddy-tracker algorithm (https://zenodo.org/records/7197432; Delepoulle et
al., 2022), based on the methods of Mason et al. (2014), Chelton et al. (2011), Kurian et al. (2011), and Penven et al. (2005).

The approach was based on the principle that, in a geostrophic regime, closed contours of SSH anomalies approximately followed streamlines. Eddy centers were identified as local extrema of ADT — maxima for anticyclonic eddies and minima for cyclonic eddies. Eddy edges were defined as the outermost closed ADT
contours corresponding to the location of maximum geostrophic velocity, i.e., where the SSH gradient was strongest (Chaigneau et al., 2008).

The algorithm identifies closed ADT contours outward from the center in 1 mm increments. A contour was considered valid if it enclosed at least 90 connected grid points, corresponding to an effective radius of ~60 km, based on the MIOST effective resolution. An amplitude threshold of 2 cm was applied to ensure the significance
of detected structures and prevent excessive detections close to the coastline. Additionally, a shape criterion was used to exclude highly deformed structures that would inhibit coherent rotation. Contours with a shape error exceeding 50% were discarded (Kurian et al., 2011; Mason et al., 2014).

### 3.2 ISW Detection

#### 3.2.1. Spectral Analysis

Each SWOT track was subdivided into several windows located before and after the interaction zone between ISWs and the targeted eddy (Table 1). At first, for each window, the mean along-track wavelength spectrum was computed from ADT_swot signal to identify dominant wavelengths (Fig. 2.A., black spectrum). In this region,





ISWs come from the IT disintegration so we consider that distance between ISW packets correspond to typical wavelengths of IT-modes (Magalhaes 2022, De Macedo et al., 2023; Tchilibou et al., 2022). Based on this, in second step, the dominant wavelengths of the spectrum (Fig 3.A, black spectrum) were isolated using a band-pass filter (Fig 3.A, blue spectrum), based on ranges corresponding to typical IT-modes: mode 1 (180–100 km), mode 2 (100–60 km), and mode 3 (60–30 km) (De Macedo et al., 2023; Tchilibou et al., 2022). The filter was constructed in the frequency domain by applying a Fast Fourier Transform (FFT), retaining only spectral components corresponding to the targeted wavelengths (Fig 3.A, blue spectrum). The filtered ADT_swot signal (Fig 3.B, blue line, bottom panel) was then reconstructed via inverse FFT. In the third step, local positive maxima in the filtered signal (Fig 3.B, blue line, bottom panel) were extracted (Fig 3.B, black crosses on blue line, bottom panel). Finally, each extracted pixels was mapped back on raw ADT_swot (Fig 3.B, black crosses on grey line bottom panel). This is done for all along track pixels of ADT_swot (Fig 3.B, black crosses on ADT swot swath, top panel). We coupled sigma 0 measurements for verify that the detected crests correspond to ISWs. Note that we tested different window sizes, which are presented in the Appendix. We specifically verified that windows smaller than 500 km provide the best correlations between the filtered signal (i.e., the detected ISWs) and the raw signal. In this study, we chose the largest possible window size to minimizing edge effects while avoiding the inclusion of additional submesoscale processes.

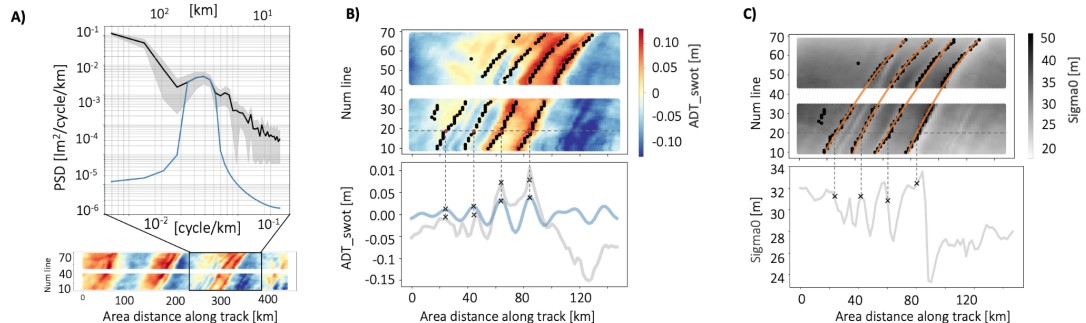

Figure 3 : Method for one ISW packet detection. A) Black line is mean along-track power spectrum density of ADT_swot signal with standard deviation envelope in gray and blue line is mean along-track power spectrum density of filtered signal B) ADT_swot. The grey line represents raw ADT_swot signal, while blue line shows signal filtered with pass-band filter between 30 km and 10 km along pixel line number 19. Filtered ADT_swot maxima are indicated by black pixels on ADT_swot. C) Sigma 0. The grey line represents sigma0 along pixel line number 19. Filtered ADT_swot maxima are shown by black pixels on sigma0. The orange line is a polynomial interpolation through these black pixels.

### 3.2.3 Polynomial interpolation

After identifying the position of ISWs occurrence (Fig 3.C. black crosses), ISWs crests were reconstructed by interpolating the local maxima using a second-degree polynomial (Fig 3.C. orange line) of the form Eq. (1)

$$f(x) = ax^2 + bx + c \tag{1}$$





The geometry of the reconstructed wavecrest was described using several parameters : the propagation axis, the concave/plane/convex geometry, the curvature intensity and the azimuth (Fig. 4).

The concave, plane, or convex nature of the crests was determined by the sign of the quadratic coefficient 'a' in the interpolation function. If 'a' was negative, the interpolation curve had a concave geometry, whereas a positive 'a' indicated a convex geometry. In Figure 4, (1),(2),(3) have a concave geometry because $a_1$, $a_2$, $a_3$ are less than 0. On the contrary, (4), (5), (6) have a convex geometry because $a_4$, $a_5$, $a_6$ are greater than 0.

Moreover, the curvature intensity was defined by the absolute value of 'a'. When 'a' was close to 0, the curve tended to be plane, and the curvature increased as 'a' deviated from 0. For example, in Figure 4 $a_4 > a_6$; hence, (6) has a lower curvature than (4).

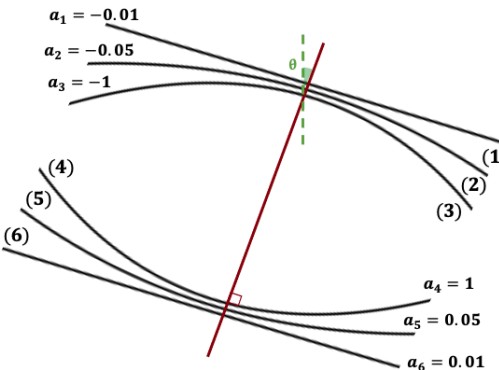

Figure 4: Schematics of wave crest characterization parameters. The red line indicates the propagation axis, $\theta$ represents the azimuth and 'a' is an indicator of curvature and geometry of crest

## 4 Results

### 4.1 Three eddy cases within the ISW propagation region

The objective of this study is to understand how mesoscale eddies influence the directional changes of ISWs propagating through the region (Fig. 5, ISWs area indicated with a gray contour). To achieve this, three representative cases were selected:

- Case 1 : No Eddy (NE) — Characterized by the absence of mesoscale eddies within the ISW propagation pathway, observed on 18 September 2023 (Fig. 5A).
- Case 2 : Cyclonic Eddy (CE) — A cyclonic eddy was present near a seamount, centered at 4.96°N, 43.08°W, on 29 September 2023 (Fig. 5B).
- Case 3 : Anticyclonic Eddy (AE) — An anticyclonic eddy, also located near the seamount, was centered at 4.11°N, 42.76°W, on 22 August 2024 (Fig. 5C).

These scenarios provide a suitable framework for analyzing the diverse interactions between ISWs and mesoscale



eddy structures, and for assessing how such interactions modulate ISWs trajectory, geometry, and propagation
behavior.

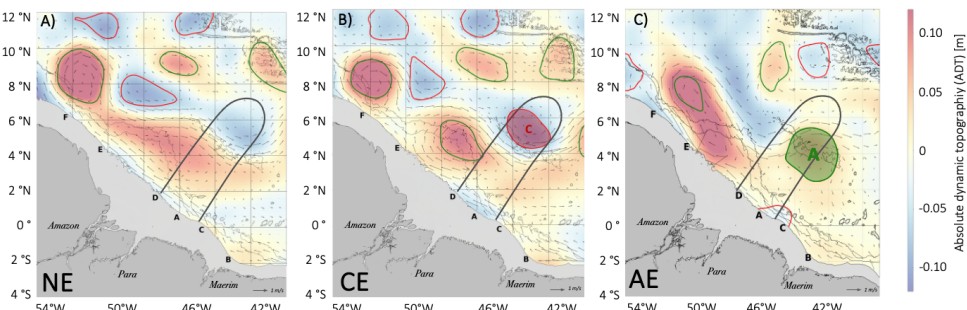

Figure 5 : Eddy detection map based on MIOST L4 filtered ADT with a 1000 km cutoff for 3 cases A) No eddies
in ISWs propagation area 18/09/2023 B) Cyclonic eddy in ISWs propagation area 29/09/2023 C) Anticyclonic
eddy in propagation area 22/08/2024. Cyclonic eddies (C) and anticyclonic eddies (A) are marked by red and green
circles, respectively. Grey contours delineate the typical area where ISWs are found propagating from sites A and
D. Black arrows represent geostrophic velocities. NE=no eddy ; CE= cyclonic eddy; AE=Anticyclonic eddy.

The three cases occur close to neap tides minima (Fig. 6). These minima are predicted by the FES2012 tidal model,
based on the dominant M2 and S2 components within the AMAZONE36 domain. During these moments, energy
levels are relatively low and comparable across cases. This configuration enables a coherent analysis of ISW
energy variations by minimizing the influence of tidal variability.

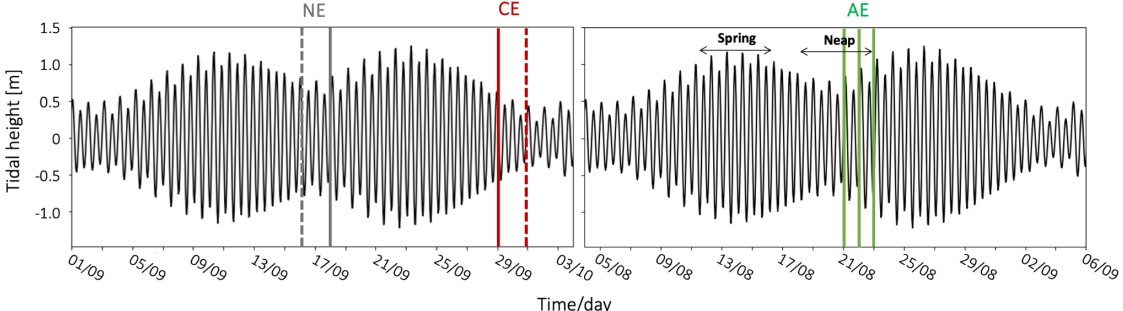

Figure 6 : Barotropic tide prediction based on M2 and S2 harmonics from FES_M2_S2 AMAZONE36. Solid lines
indicate the dates of the 3 show cases identified on SWOT and MIOST data and. A) No eddies in ISWs propagation
area 18/09/2023 B) Cyclone in ISWs propagation area 29/09/2023 C) Anticyclone in  propagation area 22/08/2024.
Dotted lines indicate the date of sunglint data acquisition. NE=no eddy ; CE= cyclonic eddy; AE=Anticyclonic
eddy

**4.2 Signature of ISW, refraction and diffraction from the interaction with eddies**



For each case it is essential to confirm that the signatures observed in SWOT (ADT_swot) are caused by ISWs. To do so, we use sunglint images with broader spatial coverage. This helps distinguish ISWs from other features such as fronts or filaments. MODIS Terra/Aqua and NOAA-20 optical images (Fig. 7A, 7B, 7C) clearly reveal a succession of crests in all three cases. These crests appear as alternating bands of increased and decreased sea surface roughness. This pattern is consistent with ISW signatures described in previous studies (Alpers, 1985; da

Silva et al., 2011; De Macedo et al., 2023). ISWs crest show spatial regularity. They repeat coherently within the region of high ISW activity (Fig 5, grey area). As observed by De Macedo et al., (2023) (Fig. 2, black dots), ISWs follow a straight path from the continental slope offshore before interacting with the mesoscale structures. In the first case (NE), SWOT and sunglint sample ISWs initially propagating from site A (Fig. 7A and Fig. 7D). In the two other cases (CE and AE), sunglint images (Fig. 7B and 7C) and SWOT data (Fig. 7E and 7F) show a

convergence of ISWs generated at sites A and D between 3°N–5°N and 44°W–46°W. These ISWs follow oblique propagation trajectories and eventually converge, forming a distinct V-geometry wave crest toward the eddy region.

From these images, we derive a key result of this study: after interacting with mesoscale eddies, ISWs follow

distinct trajectories in each of the three analyzed cases. In the NE case, sunglint (Fig. 7A) and SWOT data (Fig. 7D, pass 505) show a straight propagation path up to 8°N (Fig. 7A, orange arrow). In contrast, in the CE case (Fig. 7E, pass 227; Fig. 7B), the ISWs are refracted northwestward after interacting with the cyclone (Fig. 7B, top orange arrows). In the AE case, the ISW resulting from the convergence splits into two branches as it approaches the western edge of the anticyclone. One branch is refracted northward (Fig. 7F, pass 074), while the other is

refracted eastward (Fig. 7F, passes 046 and 018; Fig. 7C). This eastern branch appears to follow the northern edge of the anticyclone, and NECC current. Further east, near 45°W, a third refracted branch is visible, also directed northward (Fig. 7F, pass 046).The three cases offer a clear and contrasted sample of the diverse responses resulting from these complex interactions.



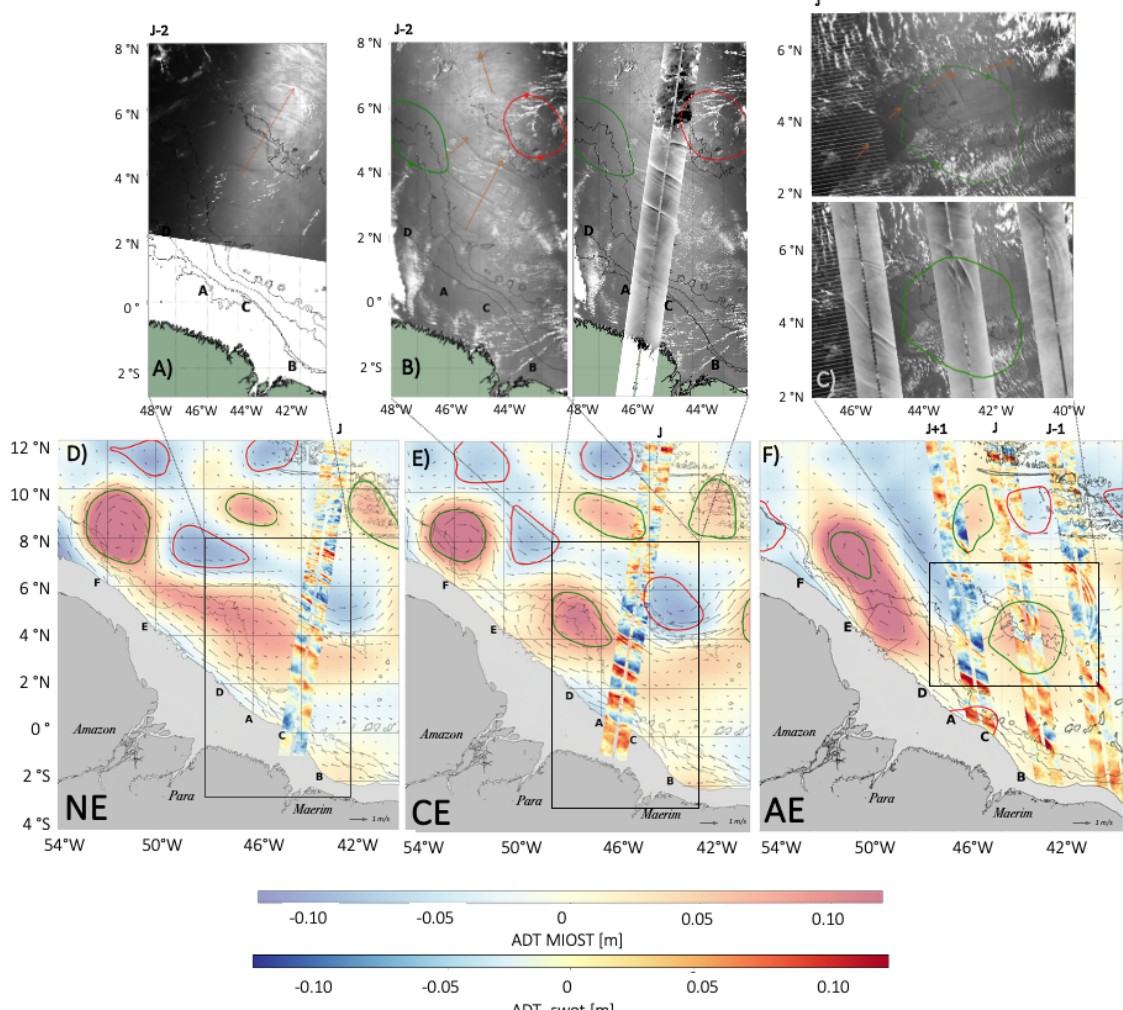

Figure 7 : Eddy detection maps based on MIOST L4 ADT (filtered with a 1000 km cutoff) and ADT_swot SWOT L3, combined with Level 1B optical imagery and SWOT sigma0 data: A) MODIS–Terra (2023/09/16) B) MODIS–Aqua (2023/10/01) C) NOAA-20 (2024/08/22) D) SWOT cycle 003, pass 505 (2023/09/18) E) SWOT cycle 020, pass 227 (2023/09/29) F) SWOT cycle 020, passes 018, 046, and 074 (2024/08/21–23). Cyclonic eddies (C) and anticyclonic eddies (A) are marked by red and green circles, respectively. NE=no eddy ; CE= cyclonic eddy; AE=Anticyclonic eddy. Bathymetry is represented using isocontours at –3500 m, –3000 m, –100 m, and 0 m

**4.3 Impact of eddies on ISW characteristics**

**4.3.1 Spectral analysis : dominant wavelength**





The extraction of ISW crests on SWOT ADT_swot field reveals the geometry of ISW and provides an initial insight into their response upon interaction with eddies. In this section, we present the results of the detection method described in section 2.2. Each track was divided into several regions associated with different dynamics before and after interaction with the eddies (Fig. 8, Table 1).


Table 1 : Characteristics of ISWs detected during (NE) 2023/09/18, (CE) 2023/09/29 and (AE) 2024/08/22

| | NE | | CE | | | | AE | | | | | |
|---|---|---|---|---|---|---|---|---|---|---|---|---|
| SWOT cycle/pass | 003/505 | | 004/227 | | | | 020/074 | | | 020/046 | | 020/018 |
| Area (°N) | 1 | 2 | 3 | 3' | 3'' | 4 | 5 | 5' | 6 | 8 | 8' | 9' |
| | 1.2-6 | 4.8-8.5 | 0.7-5.5 | 1.8-3.2 | 3-4.4 | 5.1-8.7 | -0.5-4.8 | 2.7-4.8 | 4.8-8 | 4.2-8 | 4.2-6.4 | 4.5-7.6 |
| Signal length (km) | 505 | 410 | 505 | 134 | 134 | 410 | 573 | 214 | 337 | 404 | 225 | 331 |
| Wavelenght pass-band filter (km) | 200-100 | 30-20 | 200-100 | 30-10 | 30-10 | 58-22 | 200-150 | 30-18 | 20-10 | 20-10 | 45-25 | 20-35 |
| Number of ISWs detected | 3 | 9 | 3 | 4 | 2 | 7 | 3 | 6 | 9 | 4 | 3 | 3 |
| $\bar{a}$ | -0.05 | -0.18 | -0.07 | -0.076 | -0.073 | -0.64 | -0.08 | -0.2 ; -0.01 | -0.13 | -0.03 | -0.82 | -0.740 |
| $\theta$ (°) | 27.7 | 27.6 | 29.7 | 29.7 | 27.9 | -20.4 | 25.6 | 39.8 ; 35 | -1.6 | 8.57 | 51.3 | 52.3 |

The spectral analysis of the ISWs fluxes sampled before seamount/eddy interaction area (Fig. 8A, B, C; black
spectra) shows similar patterns: each spectrum highlights two peaks at wavelengths of 170–140 km and 75 km, corresponding to modes-1 and 2 IT wavelengths, respectively.

A major finding is that after crossing the seamount and interacting with eddies, the spectral analyses of the three cases differ markedly. None of the spectra show peaks associated with mode-1 IT wavelengths between 180 km and 100 km (Fig. 8.A, B, C; red spectra). All spectra exhibit high energy levels at smaller scales. Specifically, in
the NE case, after the ISWs cross the seamount, the spectrum shows elevated energy at scales below 90 km, with peaks at 80 km, 45 km, and 25 km, indicating the presence of mode-3 IT (Fig. 8.A; red spectrum #2). For CE case, spectrum shows highers peaks between 50 and 25 km (Fig. 8.B; red spectrum #4). For AE case, the spectra of the



branches refracted to the north display generally higher energy levels at wavelengths below 30 km (Fig. 8.C; red spectra #6 and #8).

Then, the analysis was extended to characterize wave trains detected in SWOT data. In NE case, the wavelength spectra show no important peaks, indicating the absence of secondary structures near the principal wave crest in ISWs packet (Fig 8.D, black spectrum #1' and #1''). In CE case, the spectra around individual ISWs packets from A-D flux are dominated by components at 12 km ( Fig 8.E, black spectrum #3' ) and by peacks around 20 km (Fig. 8.E, black spectrum #3''). In the AE case, the spectra associated with the eastward-deflected branch (Fig. 8.F, black spectra #8' and #9') and with the main flux from A to D (Fig. 8.F, black spectrum #5') show energy peaks between

20 and 40 km in wavelength

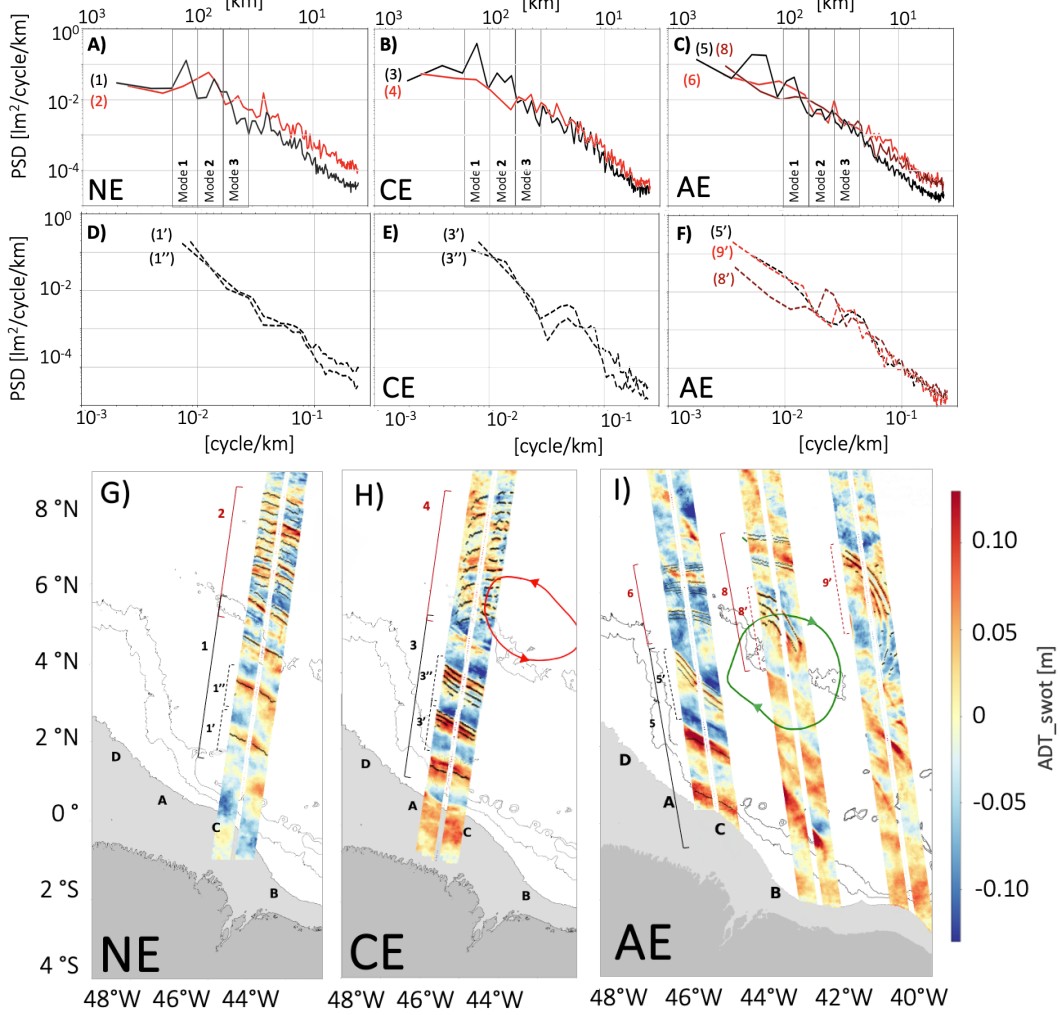



Figure 8: Mean power spectrum density of SWOT ADT_swot along track for each area for A) NE 2023/09/16 B)
CE 2023/09/29 C) AE 2024/08/22. Black (red) lines refer to  spectrum located before (after) interaction with
seamount and eddy. Dotted line refers to spectrum of single wave packet and solid line spectrum of ISWs . Area
number is indicated between parenthesis. ADT_swot with ISWs detection for  G) NE 2023/09/16 H) CE
2023/09/29 I) AE 2024/08/22   NE=no eddy ; CE= cyclonic eddy; AE=Anticyclonic eddy. Bathymetry is
represented using isocontours at –3500 m, –3000 m, –100 m, and 0 m

### 4.3.2 Wavelengths variability and ISWs-mode shifts

The wave crest detection shows that, in all three cases, the ISWs generated close to the IT source sites exhibit
inter-packet distances comparable to the wavelength of IT mode-1. (Fig. 8.G, H, I). In case CE, where the fluxes
from sites A and D converge, wave packet signatures emerge with decreasing crest-to-crest distances, ranging
from 20 km down to 15 km away from the carrier wave (Fig. 8.H, areas 3" and 3'). Similarly, in case AE (Fig.
8.I), the merging of A and D fluxes is associated with wave packets signature characterized by distances between
crest ranging from of 25–20 km (Fig. 8.I, area 5'). In contrast, no wave packet signature is observed before
seamount in case NE (Fig. 8.G).

After the interaction with eddy or seamount, the distance between wave packets differs significantly across the
three cases. In NE (Fig. 8.G), the wave crests are spaced approximately 25 km (Fig. 8.G, area 2), suggesting a
transformation from IT-mode-1 to IT mode-3. In CE, the distance  between crests in the refracted flux are shorter
than in the incident flux, around 35–40 km (Fig. 8.H, area 4). In AE, the portion of the flux that is refracted
northward shows waves packets with 10–12 km crest spacing inside each wave packet. (Fig. 8.I, area 6). Due to
data gaps between SWOT ground tracks, it is not possible to resolve the wavelengths of the flux from A that is
deflected eastward by the anticyclone. However, we identify ISWs wave packets sea surface signature. The main
wave gradually degenerates into smaller secondary waves characterized by distance between crests between 25
and 40 km (Fig. 8.I, areas 8' and 9'). Finally, the third branch, refracted northward along the edge of the
anticyclone, is characterized by a short distance between crests of 10–12 km (Fig. 8.I, area 9')

### 4.3.3 Wavecrest geometry and direction of propagation

After reconstructing the wave crests using a second-degree polynomial fit, it is observed that in the NE case, the
ISWs corresponding to IT mode-1 generated at sites A and D propagate northeastward $\theta_1 = 27.7°$ (Fig. 9, area 1,
black circles). The wavecrest has a relatively plane geometry, with an average curvature coefficient of $a_1 = -0.05$.
During the crossing of the seamount, the propagation direction remained unchanged ($\Delta\theta_{1-2} = 0$) (Fig. 9, area 2, red
circles). However, a decrease in the coefficient 'a' is observed, reaching an average of $a_2 = -0.18$ (Fig. 9, area 2,
red circles). This indicates an increase in curvature and a more pronounced concavity of the wavefront.

In the CE case, the waves sampled before interaction also propagate northeastward, with $\theta_3 = 29°$ and an average
curvature coefficient of $a_3 = -0.07$ (Fig. 9, area 3–3"–3', black triangles). These characteristics are similar to those
observed in the NE case. After interaction with the cyclone and the seamount, a significant change in propagation
direction is observed: the wavecrest is refracted by $\Delta\theta_{3-4} = 50°$ toward the west. The crests of the refracted ISWs





then propagate northeastward, with $\theta_4 = -20°$ (Fig. 9, area 4, red triangles). Compared to the reference case, this deflection cannot be attributed to bathymetric effects, supporting the hypothesis that the refraction is induced by the cyclone. Furthermore, a strong increase in curvature is measured, reaching $a_2 = -0.64$ (Fig. 9, area 4, red triangles), nearly ten times higher than that of the incident wavefront.

In the AE case, the ISWs originating from point A initially propagated northwestward with $\theta_{5\text{-east}} = 25°$. These waves encountered those from site D, which propagated at $\theta_{5'\text{-west}} = 40°$ (Fig. 9, area 5'- west, black cross). The wavecrest exhibited a relatively plane geometry ($a_5 = -0.07$) (Fig. 9, area 5, black cross. Near the eastern edge of the anticyclone, ISWs packet was refracted northward. The first three reconstructed crests were characterized by increased curvature ($a_6 = -0.23$), then, during their northward propagation, the ISWs gradually flattened, eventually exhibiting a curvature similar to that of the incident ISWs ($a_6 = -0.08$) (Fig. 9, area 6, brown cross) and an azimuth of $\theta_6 = -2°$ (Fig. 9, area 6, brown cross). According to two scenarios, if the ISW branch originates from site A, it refracts northward with $\Delta\theta_{5\text{-east-6}} = 27°$. In contrast, if it originates from site D, the refraction is stronger, with $\Delta\theta_{5\text{-west-6}} = 42°$. An other part of the incident ISWs packet was refracted eastward and propagated along the edge of the anticyclone with $\theta_{8'\&9'} = 51°$ (Fig. 9, areas 8'-9', red cross). According two scenarios, if it come from site A, ISWs was refract with $\Delta\theta_{5\text{-east-8'}} = 27°$. In contrast if ISWs provide from site D it refract with $\Delta\theta_{5\text{-west-8'}} = 12°$. The wavefronts in this region exhibited the highest curvature values, with a coefficient of $a_{8\&9} = -0.8$ (Fig. 9, areas 8'-9', red cross). Part of ISWs was refracted northward with $\Delta\theta_{8'\text{-8}} = 43,5$ (Fig. 9, area 8, red cross), and the wavefronts were characterized by a plane front $a_8 = -0.013$.

These three cases demonstrate that the ISWs originating from sites A and D were relatively plane and that the combined effects of the seamount and the refraction induced by the cyclonic and anticyclonic eddies deflected the wave trajectories and modulated the crest curvature.

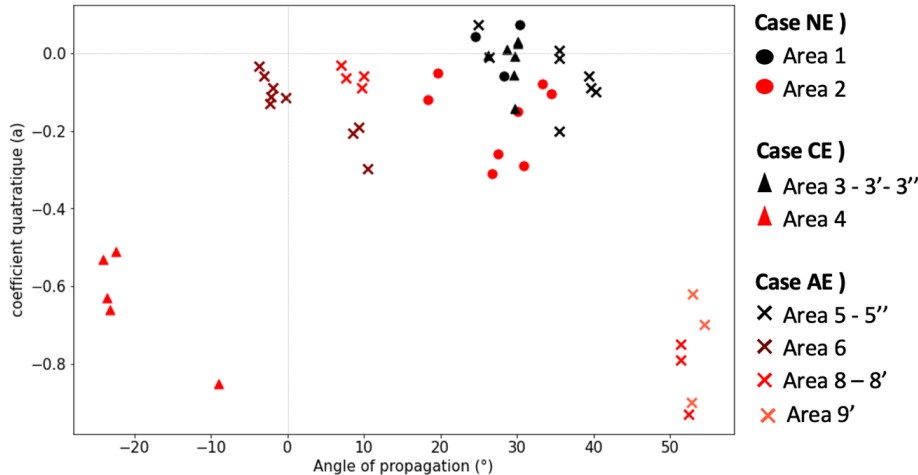


Figure 9: Orientation relative to azimuth as a function of the quadratic coefficient "a" of the ISWs wave crest detected. Each point represents an individual ISW wave crest detected in A) NE 2023/09/16 B) CE 2023/09/29 C) AE 2024/08/22. Black and red colors represent ISWs before and after interactions with seamount/eddies,





respectively. For AE case brown color indicates ISWs refracted northward, red color indicates ISWs on top off

anticyclonic eddy and salmon color indicates ISWs refract eastward

## 5. Discussion

### 5.1. Convergence of A and D fluxes favors ISW formation

#### 5.1.1. Convergence region as a mixing hotspot

In our study, we observe that the ISWs emitted from sites A and D predominantly exhibit an inter-packet spacing

characteristic of mode-1 IT and converge in both case studies (AE and CE) between 3°N–5°N and 44°W–46°W.
These results are in good agreement with modeling outcomes (Tchilibou et al., 2022 ; Kouogang et al., 2025b in
preparation) that demonstrated this convergence, as well as the study by De Macedo et al. (2023), which identified
the convergence region of these fluxes as a hotspot of intense mode-1 ISW activity. Ultimately, this convergence
region may play a role in mixing intensification, as suggested by the work of Kouogang et al. (2025a), based on

recent direct in-situ measurements from the AMAZOMIX program.

#### 5.1.2. Oblique wave-wave interaction

In the interaction region between fluxes A and D, we observe a V-geometry wave crest suggestive of an oblique
interaction in both AE and CE cases. According to the literature (Yuan et al., 2018; Yuan et al., 2023; Wang &
Pawlowicz, 2012; Helfrich et al., 2007; Shimizu & Nakayama, 2017), oblique interactions described here can

mainly be indentified in two forms :

> ➤  A merging of wave packets leading to the formation of a longer packet (Figure 10.a),
> ➤  Both solitary wave packets passing through each other without significant alteration in geometry or
>    amplitude (Figure 10.b).

In our study, for the AE case, following the oblique interaction between ISWs originating from sites A and D, the

lack of data just after the convergence point prevents us from concluding the type of interaction and the subsequent
ISW trajectories. Several scenarios are possible. It could involve a merging of wave fronts (Fig 10.a) followed by
divergence under the influence of an anticyclone: ISWs from A could be deflected eastward, while those from D
would be deflected westward. A second possible scenario in the AE case (Fig 10.b) is that ISWs from A and D
meet, cross paths, and are deflected in opposite directions—northward for those from A, and eastward for those

from D.

The different case studies in our research illustrate the complexity of possible outcomes and suggest that the
offshore Amazon region is particularly favorable for studying these complex nonlinear wave interaction





phenomena.

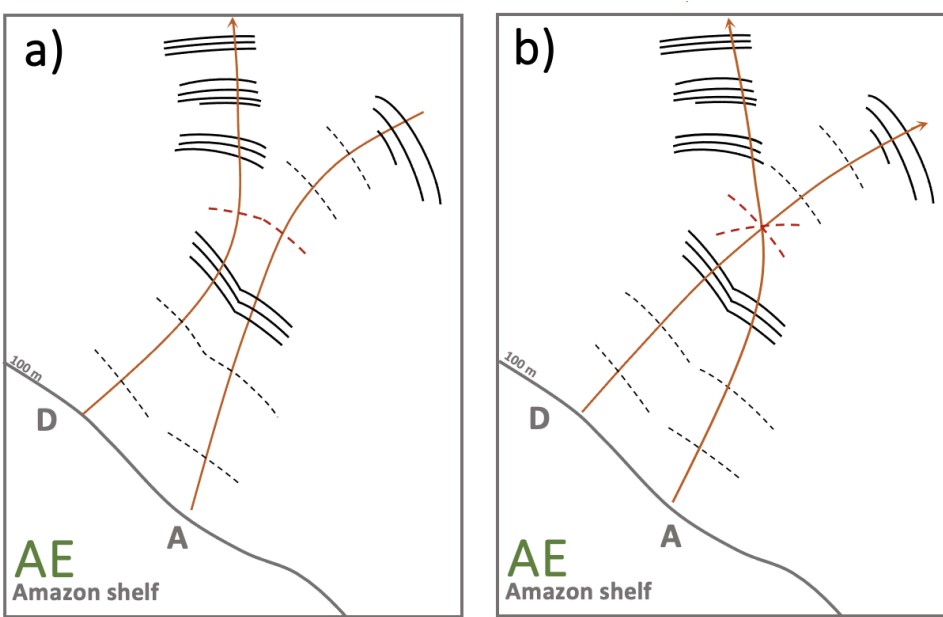

Figure 10 : Interaction between ISWs detected (black lines) off site A and D in AE case (2024/08/22) according to two propagation trajectory scenarios a) divergent trajectories with hypothetic front merge (red dot line) b) hypothetic crossing trajectories (red dot line)

**5.2. Refraction and distortion of ISWs after eddy interaction**

**5.2.3. Separating the effects of the NECC and eddies**

The NECC is closely linked to mesoscale eddy dynamics, which makes it difficult to isolate its specific impact on ISW propagation. Nevertheless, its role deserves particular attention. Indeed, in the AE case, ISWs appear to be deflected eastward toward the edge of the eddy prior to the interaction, following the NECC streamlines. This intense zonal current, with a variable path, could indeed influence ISW trajectories. This observation supports the hypothesis proposed by Tchilibou et al. (2022), De Macedo et al. (2023), Magalhães et al. (2016) and Kouogang

et al., (2025b, in preparation) suggesting that the strengthening of the NECC plays an important role in the acceleration, refraction, diffraction and shift ITs associated to ISWs in the northeastern region. Recently numerical investigations showed that internal tide–eddy interactions in the region lead to diffraction at the edges of eddies, and to refraction near the eddy cores. The realistic simulations also revealed an instantaneous splitting of the IT energy fluxes caused by shear instabilities associated with the NECC (Kouogang et al. 2025b, 2025c, in

preparation). For comparison, it has been observed that the meandering Gulf Stream significantly refracts and traps ITs (Duda et al., 2018), highlighting the ability of such a current to act as a barrier or waveguide for ITs. By analogy, the contribution of the NECC in the deflection or distortion of ISW crests cannot be overlooked using



satellite observation, and a dedicated analysis would be necessary to assess its relative contribution using idealized numerical experiments.

**5.3. Distortion of wave crests**

This study shows that ISW deflections substantially increase wave crest curvature, in agreement with idealized simulations investigated in the region. (Kouogang et al., 2025c, in preparation). This is particularly evident in the AE case, when the center of the ISW front is aligned with the current along the northern edge of the anticyclone (Fig 11.c), and in the CE case, when the ISW tip is opposed to the current along the western edge of the cyclone

(Fig 11.b). Eddies feature strong gradients in both velocity and stratification, with maximum speed at the edges and minimum at the center. When ISWs interact with this spatially varying current field, their local phase speed can be altered (Lamb, 2014; Dunphy & Lamb, 2014). Some sections of the wave front may be "accelerated" where the current aligns with the direction of ISWs propagation (Fig 11.c), while others may be "slowed down" in the presence of opposing or weaker currents (Fig 11.b). This spatial variation in ISW phase speed leads to significant

distortion of the wave front, which becomes markedly more curved than the initially planar incident front.

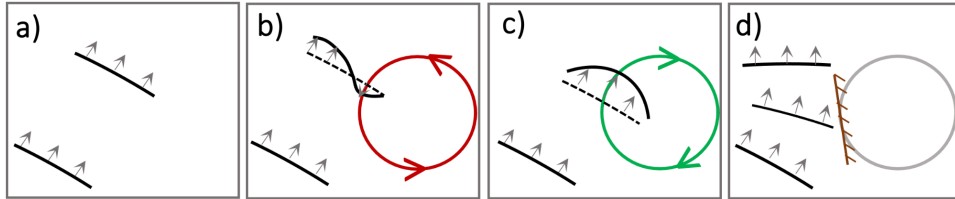

Figure 11 : Impact of eddy edge currents on the curvature of internal wave crests. Cyclonic eddies and anticyclonic eddies are marked by red and green circles, respectively.

In contrast, in the AE case, after interaction, part of the ISW flux is refracted northward (Fig 8.i). This portion of

the flux maintains a relatively flat curvature throughout its propagation. One possible explanation for the absence of wavefront distortion after refraction is that the ISWs do not interact directly with the intense edge currents of the eddy, but are instead reflected upon encountering a kind of "physical wall" (Fig 11. d). This interpretation is supported by recent studies (Guo et al., 2023; Wang & Legg, 2023 ; Kouogang et al., 2025c, in preparation), which show that stratification changes induced by eddies can refract ISWs.

This observation contributes to an ongoing debate within the scientific community regarding the relative roles of stratification and currents in the processes of internal tide refraction and ISW distortion. Several studies (Bendinger etal., 2025 ; Guo et al., 2023; Wang & Legg, 2023; Xu et al., 2024) suggest that eddy-related currents play the dominant role in ISW refraction, relegating the effect of local stratification variations to a secondary role. However, in other oceanic contexts, it is precisely these stratification changes that appear to be the main driver of internal

tide refraction and coherence loss, surpassing the influence of relative vorticity gradients and subtidal currents (Zaron & Egbert, 2014). These findings highlight the importance of regional conditions and the complex interaction between stratification and currents in shaping ISW trajectories and behavior.

**5.5. Satellite Data Limitations**



### 5.5.1. Uncertainty in the spatial and temporal positioning of eddies

The scenarios explored underscore the need for accurate estimates of eddy locations to assess their influence on ISW dynamics. However, uncertainty exists regarding the precise positioning of these eddies. They are identified from daily altimetric maps that combine measurements from several satellites passing at different times. The interpolation of these data inevitably leads to smoothing of small scales and a loss of spatial and temporal resolution. Consequently, when compared with instantaneous measurements from SWOT and MODIS/NOAA20,
it becomes difficult to determine whether the ISW–eddy interaction occurs at the eddy core or periphery. This uncertainty can limit the interpretation of observations.

### 5.5.2. Satellite Sampling

In the AE case, the lack of data between SWOT tracks prevents observation of the wavefront evolution after the wave–wave interaction. This makes it impossible to precisely characterize the type of wave–wave interaction and
also limits observations of wave–eddy interactions.

More generally, several factors hinder the observation of wave–eddy interactions: the temporal resolution of satellite sensors, environmental conditions (cloud cover, wind, ocean surface state, sun angle), and the viewing angle. These limitations also affect the interpretation of physical processes, including ISW refraction and the detection of ISW packets. In unfavorable conditions, only the leading wave with strong contrast is visible in
sunglint or $\sigma^0$ images.

Additionally, what is interpreted in the CE case as westward refraction of the wave flux by the cyclone may in fact only represent part of a diffracted wave flux, of which only the western branch is captured. Other branches to the east may exist but are unfortunately not captured in our case studies. Therefore, we cannot conclude on the full extent of the flux, as satellite images do not capture it in its entirety. This limited ISW visibility may lead to an
underestimation of the extent and complexity of eddy-induced ISW modifications. These observational limitations emphasize the need to complement the analysis with in situ measurements or high-resolution modeling to better capture the full scale and dynamics of ISWs and their interactions with eddies.

### 5.4. Inter-packet Distance and Mode Transfer

### 5.4.1. Effect of the Seamount

In the NE case, an energy transfer from mode-1IT to higher IT modes, particularly mode-3, was observed. This phenomenon aligns with numerous studies showing that steep bathymetry can disperse internal tide energy into higher modes and enhance mixing (Johnston & Merryfield, 2003; Johnston et al., 2003; Mathur et al., 2014). One hypothesis is that the presence of a seamount can locally alter stratification and the effective water column depth, thus affecting internal wave phase speed and reducing wavelength. This has been reproduced in the region using
numerical investigations (Kouogang et al., 2025b, in preparation). A broader analysis of SWOT data would be valuable to systematically confirm the impact of topography on ISW behavior.

### 5.4.2. Combined Effect of the Seamount and Eddies



In addition to the effect of seamounts, a similar mode transfer from mode-1 to mode-3 was observed during wave deflection by a cyclone. These findings are consistent with earlier studies (Guo et al., 2023; Dunphy & Lamb, 520 2014; Kouogang et al., 2025b, in preparation), which demonstrated that a mode-1 internal tide interacting with an eddy can transfer energy to higher modes, thus reducing the main mode's energy flux. Moreover, in the AE case, we observe the emergence of wave trains (three crests following the main one, with inter-packet distances on the order of 10 km) in the deviated branch after interaction with the anticyclone. In contrast, in the NE case, the main crest appears alone, i.e., without a trailing wave train.
Our results raise the hypothesis that the eddy may destabilize the ISW's main crest and enhance wave train formation. More quantitative studies are needed to refine this hypothesis.

**5.5. Detection Method and Perspectives**

The method used in this study to detect ISW wavefronts is innovative and would benefit from automation. However, this approach presents several limitations that currently prevent its generalization. It is particularly 530 sensitive to the choice of analysis window size, changes in wavelength, and the presence of submesoscale processes also visible in SSH, which remain difficult to automatically separate. Additionally, the choice of filtering window size affects detection precision. Nevertheless, characterizing ISWs based on their morphology offers a promising avenue for detecting and isolating these structures in SSH fields observed by SWOT, especially when coupled with $\sigma^0$ measurements.

**6. Conclusion**

This study investigates the impact of mesoscale eddies on the characteristics of ISWs off the Amazon Shelf, focusing on their distance between crests, mode, propagation direction, and crest curvature. The analysis is based on the extraction of ISW signals from SWOT L3 KaRIn wide-swaths measurements, and the identification of eddies from daily MIOST ADT maps. Three distinct interaction scenarios were analyzed: a case of propagation 540 without interaction, a case of refraction by a cyclonic eddy, and a case of diffraction by an anticyclonic eddy.

It was shown that, prior to any interaction with eddies or bathymetric features, the ISWs generated from ITs whose origins are sites A and D propagated with angles ranging from 25° to 28° relative to the north-south axis. The wavefronts exhibited plane geometries and were dominated by crest spacing corresponding to wavelength of IT mode-1. The presence of a seamount did not affect ISWs propagation but induced a shift toward crest spacings 545 characteristic of IT mode-3 (Fig. 12A). In contrast, interaction with the western edge of a cyclonic eddy and seamount resulted in a 50° westward refraction of the wave train. This interaction was accompanied by a significant increase in crest curvature and a reduction in inter-packet distance, indicating a shift toward crest-to-crest distances consistent with IT mode-3 (Fig. 12B). Finally, interaction with the western edge of an anticyclonic eddy and seamount led to diffraction. One branch of ISWS refracted westward, exhibiting flatter wave crests and several 550 waves packets. Simultaneously the other ISWs branch was deflected eastward, with the crests becoming highly curved and wave packets emerging (Fig. 12C)

This study provides the first observational evidence of ISWs refraction and diffraction after ISWs interact with eddies of different polarity. The detection method developed in this study proved promising in highlighting the



diversity of ISW responses regarding eddy structures and location with respect to the ISW path. As a continuation,
applying this approach to other regions and case studies would be valuable in broadening our understanding of
ISW/eddy interaction variability. The 250m SWOT data might also be used to reveal other fine scales features of
the ISW and their interactions with eddies. A comparison with results from idealized or non-hydrostatic 3D-models
would also help clarify the respective roles of eddies, background currents, and stratification in shaping ISW
dynamics.

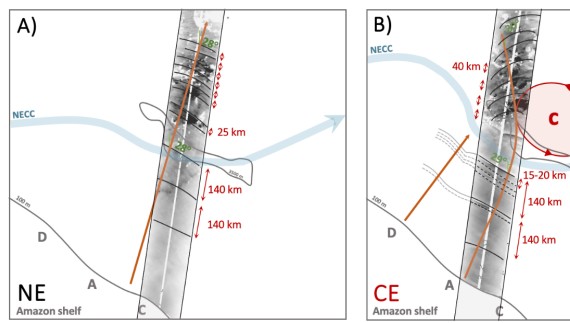
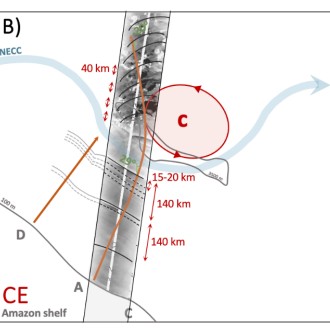
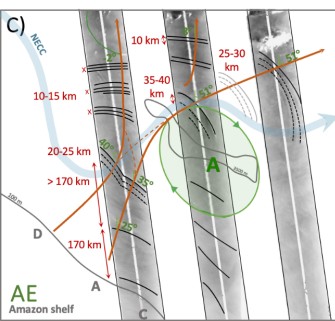


Figure 12 : Interaction between ISWs detected (black lines) off site A and D for A) NE 2023/09/16 B) CE
2023/09/29 C) AE 2024/08/22 on sigma0 SWOT data. Grey lines denote ISWs visible on MODIS and NOAA-21
sunglint images. Orange line denotes axis of propagation.

**Data availability**

The SWOT L3_LR_SSH product, derived from the L2 SWOT KaRIn low rate ocean data products (NASA/JPL
and CNES), is produced and made freely available by AVISO and DUACS teams as part of the DESMOS Science
Team project". AVISO/DUACS, 2024. SWOT Level-3 KaRIn Low Rate SSH Expert (v2.0.1). CNES.
https://doi.org/10.24400/527896/A01-2023.018


DT merged all satellites Global Ocean Gridded Experimental SSALTO/DUACS Sea Surface Height L4 product
and derived variables are available by AVISO and DUACS teams. These products were processed by
SSALTO/DUACS and distributed by AVISO (https://www.aviso.altimetry.fr) supported by CNES. DOI:
10.24400/527896/a01-2004.007


Level 1B MODIS/TERRA (doi: 10.5067/MODIS/MYD021KM.061), and NOAA20 (doi:
10.5067/VIIRS/VJ102MOD.021) data were collected from NASA's Earth Science Data System, ESDS
(https://earthdata.nasa.gov/)

**Authors contributions**

AKL supervised the overall study and provided scientific guidance throughout the work and financial support CG
(voir fabius ou carina), CG make analysis and writing. FK and AKL contributed through regular discussions and
technical assistance. AKL, JD and JM contributed to the interpretation of results, and identified the AE case. CD
processed and provided MODIS TERRA/AQUA and NOAA-20 satellite data. AKL, CA,MT, ID, and SB provided



support for spectral analysis and signal processing. AD provides a py-eddy-tracker algorithm. MB supplied and supported the use of MIOST L4 data. AH and LC help for all discussion. CG wrote the paper with contributions from all co-authors.

**Competing interests**

The authors declare that they have no conflict of interest.

**Acknowledgments**

The authors would like to thank the AVISO + (Archivage, Validation et Interprétation des données des Satellites Océanographiques) and CLS ( Collecte Localisation Satellites ) team for their support and expertise in the

distribution of the data. The authors would like to thank the NASA's Earth Science Data System, ESDS for providing the MODIS/TERRA data. This work is a contribution to the project "MIAMAZ-ETI" (Multi-Sensors study of the fine scale processes and their impacts on ocean color, off the Amazon shelf : Eddy-Tides Interactions).

**Financial support**

This work is supported by CNES funding in the frame of the APR MIAMAZ-ETI project (Pis : A. Koch-Larrouy, C. Artana, I. Dadou)

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

**Appendices**

**Sensitivity tests**

Each SWOT track was subdivided into several windows. In this appendix, we assess the sensitivity of ISW
detection to the window size (ranging from 200 km to 850 km). For demonstration purposes, we focus on mode-1 ISWs, though the results are similar for other wavelengths. Figure A1 illustrates the results of these tests for pass 227, cycle 004.

For all window lengths, the associated spectra show a peak in the 100–200 km band (Fig. A1.A), corresponding to mode-1 internal tides/solitary waves. This peak is clearly visible for all window lengths, except perhaps for the
275 km window (Fig. A1.A.a), which truncates the spectral intensity. We conclude that window lengths greater than 275 km provide sufficient spectral resolution to isolate the structures of interest.

The identification of ISW crest positions projected onto the SWOT tracks shows that for windows between 275 and 505 km (Fig. A1.B.a,b,c), there is a high correlation between the raw and filtered signal (r > 0.6, Fig.



A1.C.a,b,c), with a low standard deviation ($0.01 < \sigma < 0.1$, Fig. A1.C.a,b,c) along the entire track. In contrast, for
longer windows (620 to 845 km, Fig. A1.B.d,e,f), the correlation drops beyond 500 km ($r < 0.6$, Fig. A1.C.d,e,f),
and the standard deviation increases ($0.01 < \sigma < 0.3$, Fig. A1.C.d,e,f). This drop in correlation is likely related to
additional high-frequency submesoscale processes (Fig. A1.B.d,e,f, grey curve).

Furthermore, for the smallest window (275 km, Fig. A1.A.a), the correlation over the first 100 km is $r = 0.85 \pm
0.03$ (Fig. A1.C.a), while for the longest window (845 km), it reaches $r = 0.90 \pm 0.02$. This is likely due to edge
effects caused by the implicit periodicity assumption of the Fourier transform. These effects are limited because
the start of the window coincides with the beginning of the ISWs. Additionally, although spectral truncation can
theoretically produce edge artifacts (Gibbs phenomenon), no such artifacts were observed in our tests.

We conclude that, in this case, windows between 275 and 505 km (Fig. A1.a,b,c) allow for proper extraction of
ISW crests, with limited edge effects. We select the largest valid window (505 km) as it offers the best
compromise—minimizing edge effects while avoiding the inclusion of additional submesoscale processes. This
choice is also supported by theoretical considerations: according to Oppenheim et al. (2010), reliable spectral
analysis requires the signal length to be at least twice the target wavelength.



A)

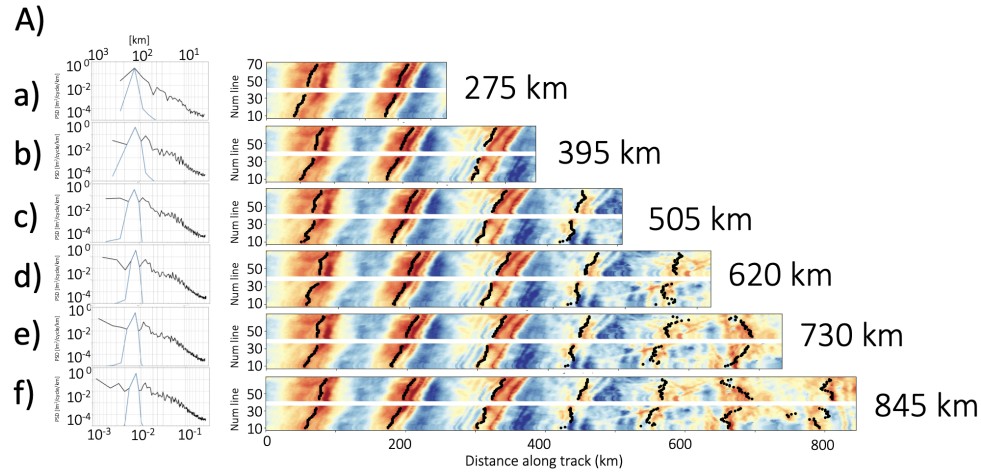

B)

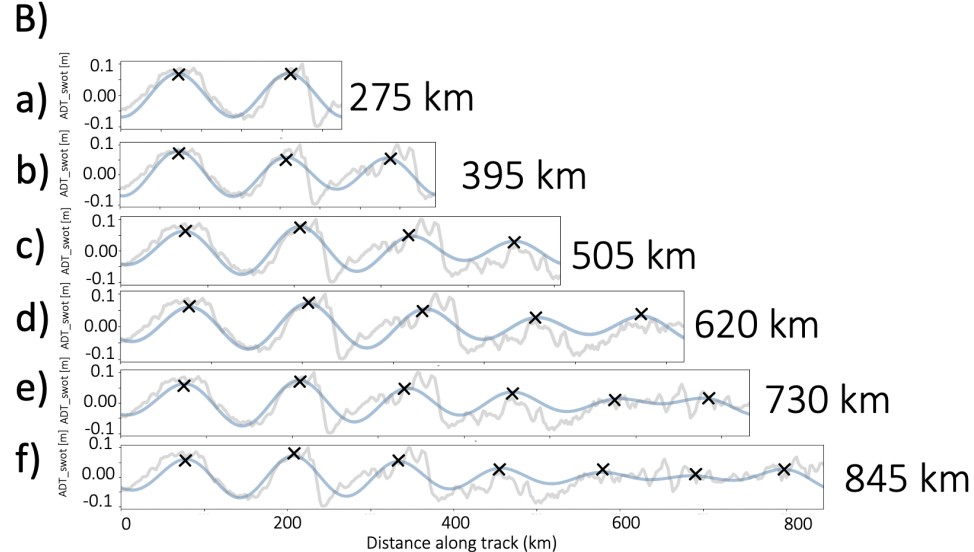




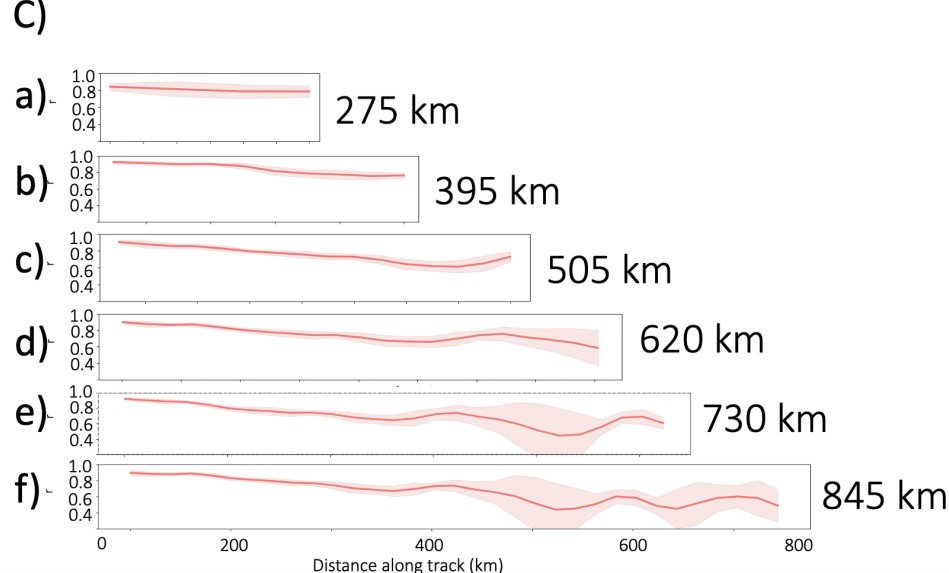

Figure A1 : A) Mean power spectral density of SWOT ADT_swot along-track and ADT_swot with ISWs detection for 6 signal lengths from 200 km to 850km. B) The grey line represents the raw ADT_swot signal, while the blue line shows the signal filtered with a pass-band filter between 200 km and 100 km along pixel line number 53. C) Mean Pearson correlation between the raw ADT_swot signal and the filtered signal.