# Peer review of "Internal solitary waves refraction and diffraction from interaction with eddies off the Amazon Shelf from SWOT"

_EGUsphere, 2025_

## Referee Comment (RC2)

**Review of**

**Internal solitary waves refraction and diffraction from interaction with eddies off the Amazon Shelf from SWOT**

October 8, 2025

The authors explore internal solitary waves (ISWs) off the Amazon Shelf as measured by the recently launched SWOT satellite altimetry mission, complemented by sunglint images. The novelty of this study lies in the observational evidence of ISW refraction/diffraction due to interactions with mesoscale eddies of different polarity. To do so, the authors first extract ISWs from SWOT swaths through spatial bandpass-filtering technique knowing the typical length-scale of ISWs in the region from previous studies. The extracted ISWs are then quantified including a characterization of their geometry (propagation axis, concave/plane/convex wavecrest, curvature intensity, azimuth). These characteristics are classified by investigating three different case studies with each case study corresponding to a different background mesoscale eddy field: No Eddy (NE), Cyclonic Eddy (CE), Anticyclonic Eddy (AE).

The presented work is a valuable addition to the assessment of SWOT and the understanding of ISW and their interaction with eddies. Overall, the manuscript is well written and easy to read. I recommend its publication in Ocean Science. I do have some comments, which I consider mostly minor, listed below.

Comment #1 One concern that I have had is how the background conditions are interpreted. The NE and CE case appear to represent the same mesoscale background field (NE: 18 Sep 2023; CE: 29 Sep 2023). In other words, the cyclone and anticyclone detected in the CE case are also evident in the NE case. Just because the eddy tracking does not identify those as eddies, the ADT and associated currents highly in Figure 5a and 5b suggest eddy or mesoscale activity, don't they? Are the associated currents in the NE case significantly weaker than in the CE case? I guess that for the given day (18 Sep 2023) there are no enclosed ADT contours for the anticyclone and cyclone in ISW propagation direction? The eddy detection considers only local extrema of ADT? Is it possible to add the velocity field (local normalized angular momentum) as it is done in the AMEDA algorithm (Le Vu et al., 2018). I assume that the authors have not found a better example for NE with corresponding SWOT passes. The authors may stick with the given example but change the definition of NE, which for now is defined as "absence of mesoscale eddies" to something like "absence of interaction...".

Comment #2 Following up the above a comment, there is a lot of speculation on the ISW propagation direction after encountering seamounts and eddies, though this is well discussed and limitations of the presented manuscript are clearly listed (see Discussion). A ray tracing experiment that models the horizontal propagation of tidal rays (separated by vertical modes) through a mesoscale eddy field (see Rainville and Pinkel, 2006; Vic et al., 2023; Bendinger et al., 2024 or a more dedicated approach by Duda et al., 2018, Guo et al., 2023) would certainly remove any doubt on the ISW propagation direction proofing whether the observed mesoscale eddies (CE and AE) explain indeed the observed ISW refraction in propagation direction and whether NE causes no refraction in propagation direction. Such ray tracing could be applied to the three surface geostrophic velocity fields in Figure 7D to 7F (or Figure 12) to see whether it matches the changing propagation of ISWs observed in SWOT and sunglint images. However, I understand that this involves further work, which is why I leave it to the authors whether they want to quantify their findings with theory.

Comment #3 Before focusing on how eddies impact ISW characteristics, I would start with a small paragraph describing their mean characteristics (based on Table 1). How many ISW were detected, etc.? I think it is also worth to quantify their SSH signature (>10 cm)?

**Comments by line**

lines 24-26: Personally, I think that the characterization of ISWs using SWOT deserves a bit more attention in the abstract (and in the conclusion also), stating at least how many ISW you detected and what their SSH amplitude is.

line 41: in a stratified ocean

line 62 and throughout the manuscript: To be double checked with the Copernicus editorial service whether work not submitted/published should be mentioned as reference. If so, it would be great to show associated results of those studies in preparation wherever they are relevant in the given manuscript. This could be in form of appendix, etc.

line 77 and 81: The authors introduce the terms coherent and incoherent. Briefly define what coherent and incoherent means for readers which are not familiar.

Figure 1 and 2: Figure 1 and 2 could be merged to one figure with two subplots?

line 129: MIOST maps include SWOT KaRIn and nadir observations. Do the authors expect that submesoscale and wave-like motion is contained in ADT (when SWOT swaths are available and used for the multiscale, multivariate mapping), which was used to apply the mesoscale eddy tracking? In other words, when studying mesoscale dynamics should one not exclude the SWOT KaRIN observations? I think that there is a dataset provided by AVISO/CMEMS which provides ADT maps but without SWOT. I do not expect big differences for the final eddy tracking, but this contrasts with what was being sad in lines 91-94, i.e. SWOT should be used with caution when studying mesoscale and geostrophic currents. However, in line 138 it is implied that ADT maps consist of nadir-pointing data only. Please clarify how SWOT data (KaRIn and nadir) is implemented in ADT maps.

Figure 2: It seems like, transparency has been added to the colorbar, but the shading in the actual plot has no transparency? I could be a nice addition to plot all ISWs identified in this manuscript on top of those from Macedo et al. (2023)?

line 145: across the track?

Section 2.2: I assume that the authors used the 250 m (unfiltered) product? Please clarify. If yes, also mention in the introduction in line 87 that observations are at disposal down to 250 m. Did the authors compare the 250 m and 2 km products? If yes, are there big differences between these two? I expect that the ISWs might have a stronger (unsmoothed) signature in the 250 m product?

line 158: Specifically state what is included in the high-resolution signal, e.g. tidal and non-tidal IGWs, submesoscale, etc.

line 181-183: Is this commonly done when applying eddy tracking in near-equatorial regions? Why not using SLA maps when focusing on mesoscale eddies?

line 202: Fig. 3A?

**Figure 3:** I would enlarge the figure/subplots for the better visibility.

**Figure 6:** I leave it to the authors whether they want to keep this figure. For me the text is sufficient.

Section 4.3: ISW propagation is interpreted using surface geostrophic velocities, mainly representative of mode-1. What role could higher-baroclinic current velocities play (see Duda et al., 2020, Guo et al., 2023)?

Section 4.3.1: ISWs are highly anisotropic. The significance/meaningfulness of (along-track) wavenumber spectra depend on whether the selected tracks are aligned with the primary propagation direction. The latter seems mostly to be the case. Whatsoever, I believe that this is worth mentioning. SWOT cycle 20, for the most easterly pass it could be possible that the tracks used for the spectral analysis are partly aligned with the wavecrests?

Caption Figure 8: Subplots D-F are not mentioned.

Section 5.2.3: Important section discussing the NECC. Why is there a big interest in separating effects of NECC and eddies? Does it matter whether the NECC or eddies refract/diffract ISWs? A ray tracing experiment could possibly separate these two effects when modeling the propagation of tidal rays in climatological/annual mean background current

fields (which represent the NECC?) and daily ADT/SLA maps (which represent eddies).

lines 446-450: Following a comment from above, it would be very helpful to show something from Kouogang et al. (2025b, 2025c in preparation).

lines 450: Duda et al. (2018) is a modeling study, isn't it? In that case prevent using "observed".

lines 552-553: Not as detailed as in the presented study, but I think it is worth mentioning the studies Xie et al. (2015), Xu et al. (2020), and Huang et al. (2024) who simultaneously observed ISWs and mesoscale eddies. These references might be also added to the introduction.

lines 630-632: There is now a peer-reviewed version: https://doi.org/10.5194/os-21-1943-2025

lines 650-653: Remove the doi of the preprint.

lines: 715-719: To be double checked with the Copernicus editorial service whether work which is not submitted/published should be in the reference list.

**References**

Bendinger, A., Cravatte, S., Gourdeau, L., Rainville, L., Vic, C., Sérazin, G., Durand, F., Marin, F., and Fuda, J.-L.: Internal-tide vertical structure and steric sea surface height signature south of New Caledonia revealed by glider observations, Ocean Sci., 20, 945–964, https://doi.org/10.5194/os-20-945-2024, 2024.

Huang, H., S. Qiu, Z. Zeng, P. Song, J. Guo, and X. Chen, 2024: Modulation of Internal Solitary Waves by One Mesoscale Eddy Pair West of the Luzon Strait, J. Phys. Oceanogr., 54, 2133–2152, https://doi.org/10.1175/JPO-D-23-0244.1, 2024.

Le Vu, B., Stegner, A., and Arsouze, T.: Angular momentum eddy detection and tracking 1029 algorithm (AMEDA) and its application to coastal eddy formation, Journal of Atmospheric and 1030 Oceanic Technology, 35 (4), 739–762, https://doi.org/10.1175/JTECH-D-17-0010.1, 2018.

Vic, C. and Ferron, B.: Observed structure of an internal tide beam over the Mid-Atlantic Ridge, Journal of Geophysical Research: Oceans, 128, e2022JC019509, https://doi.org/10.1029/2022JC019509, 2023.

Xie, J., Y. He, Z. Chen, J. Xu, and S. Cai: Simulations of Internal Solitary Wave Interactions with Mesoscale Eddies in the Northeastern South China Sea, J. Phys. Oceanogr., 45, 2959–2978, https://doi.org/10.1175/JPO-D-15-0029.1, 2015.

Xu, J., He, Y., Chen, Z., Zhan, H., Wu, Y., Xie, J., Shang, X., Ning, D., Fang, W., Cai, S.: Observations of different effects of an anti-cyclonic eddy on internal solitary waves in the South China Sea, Progress in Oceanography, 188, 102422, https://doi.org/10.1016/j.pocean.2020.102422, 2020.

---

## Author Comment (AC1)

**Subject: Response to Anonymous Referee #1**

Dear Madam/Sir,

We sincerely thank you for taking the time to review our manuscript,*"Internal Solitary Waves Refraction and Diffraction from Interaction with Eddies off the Amazon Shelf from SWOT"* and to provide your comments and suggestions for enhancing the quality of this paper. Below, we present a detailed, point-by-point response.

**Response to Anonymous Referee #1**

**Comment 1**: « My major concern is about the wavenumber spectra shown in Figure 8 and corresponding descriptions, especially for the internal wave modal content. According to my understanding, mode-3 internal tides (ITs) off the Amazon Shelf may have a horizontal wavelength of approximately 50km. However, it cannot be concluded that any signal with a horizontal wavelength of approximately 50 km corresponds to mode-3 ITs (Line 331). Actually, comparing spectra #2, #3 and #4 as well as the ADT_swot snapshots in Figures 8G and 8H, the peak appearing at approximately 50 km on spectrum #2 might be ISW packets rather than mode-3 ITs. »

**Response 1: T**hank you for your comment, but I am not sure I fully understand the meaning of your question.

If your comment refers to ISW packets as "intra-packet" distances (i.e., the distances between crests within the same wave packet), Macedo et al. (2023) showed that ISW packets are characterized by intra-packet distances typically ranging between 10 and 20 km (**Figure a**), which is smaller than the typical wavelengths of mode-3 waves. However, it is indeed more difficult to distinguish, based solely on the spectrum, between mode-4 or mode-5 ISWs and these intra-packet distances. To do so, it is necessary to examine the main solitary wave of the ISW packet.In addition, we produced a 1D along-track transect of the SWOT swath for the NE case (**Figure b**). Downstream of the seamount, we observe peaks separated by approximately 25 km. These crests are independent from one another and do not display the morphology of an ISW packet (i.e., a leading large-amplitude crest followed by secondary, decaying waves).

If, on the other hand, your question concerns the distinction between mode-3 ISWs and mode-3 internal tides (ITs), at our current level of understanding it is difficult to reliably separate the two. The most plausible hypothesis is that the mode-3 signals we observe correspond to ISWs carried by an internal tide through an instability of the IT's leading crest. This issue lies beyond the central questions addressed

in this study; therefore, for clarity, we use the expression "mode-3 ITs" to collectively refer to both mode-3 ITs and mode-3 ISWs observed in the data.

[Figure]

***Figure a.*** *Figure from de Macedo et al, 2023: ISW intra-packet distance distribution.*

[Figure]

***Figure b.*** *SWOT ADT_swot in the NE case (bottom panel). The grey line represents the 1D section shown in the top panel, corresponding to the ADT_swot along-track signal at pixel line number 52. The grey line shows the raw ADT_swot signal, while the blue line represents the signal filtered with a band-pass filter: 200–100 km between 0 and 505 km, and 30–20 km between 400 and 900 km.*
* * *
**Comment 2:** « Moreover, the authors mentioned in section 5.4.1 that the mode-3 ITs are reproduced in numerical investigations (Kouogang et al., 2025b, in preparation). I think that it is necessary to show some key results that support the generation of mode-3 ITs in this study or in a supporting material. »

**Response 2:** We appreciate the reviewer's interest in the comparison with the results of Kouogang et al., 2025b. However, these results are part of a separate study that will be submitted before the end of the year. The scattering processes that might occur in our study have been largely described by other studies. For example, Johnston & Merryfield, 2003; Johnston et al., 2003; Mathur et al., 2014. If the kouogang's papers is in preprint before this manuscript passes the review process I will cite it. Otherwise I will remove this reference to this work because other studies already tackle this issue.
* * *
**Comment 3:** « My second concern is also related to the no-eddy (NE) case. In both the anticyclonic eddy (AE) and cyclonic eddy (CE) cases, ISW packets are observed prior to encountering the mesoscale eddy/seamount. In the NE case, however, only isolated ISWs are observed before reaching the seamount. To draw robust conclusions regarding the influence of mesoscale eddies on ISWs, it is recommended to identify an example where ISW packets exist before the seamount interaction for the NE case. »

**Response 3:** We agree that having an NE example with ISW packets before the seamount interaction would strengthen the comparison.We have identified a new, more robust showcase show in **Figure c** (SWOT pass 227, cycle 012, which samples the ISW fluxes generated at sites A on 2024-03-14) that leads to the same results as in the previous case and in addition shows ISWs packet before any interaction. These figure is now described section 'Signature of ISW, refraction and diffraction from the interaction with eddies' (section 4.2) lines 296 -330.

So the topography and the seamount are not indispensable for the formation of ISWs packets in this case even if they migth contribute to them (Johnston & Merrifield, 2003; Lamb, 2004; Mathur et al., 2014 ). This observation is well aligned with the hypothesis formulated in Kouogang et al. (2025), which shows increased mixing at station ST14 in the AMAZOMIX data—upstream of the seamount—that the authors suggest may result from wave–wave interactions. It is also consistent with the findings of Solano et al. (2023), who show that the interference pattern between mode-1 and mode-2 waves induces nonlinear energy transfers less than 400 km from the slope.

[Figure]

***Figure c.*** *Eddy detection maps based on MIOST L4 ADT and ADT_swot SWOT cycle 012, passes 227, from 2024-03-14 combined with Level 1B optical imagery MODIS-Aqua from 2024-03-10. Cyclonic eddies and anticyclonic eddies are marked by red and green circles, respectively. NE=no interaction eddy case . Bathymetry is represented using isocontours at −400 m, −3000 m, −100 m, and 0 m*
* * *
**Comment 4:** « Figures 7A-C. It is recommended that the authors present these figures using the same spatial domain, if possible. »

**Response 4:** Thank you for this helpful suggestion. Unfortunately, it is not possible to present Figures 7A–C with the same spatial domain because they correspond to sunglint images acquired at different times by different satellites (MODIS Terra and MODIS Aqua ). We used multiple satellites because sunglint observations are strongly constrained by the factors : (1) few usable MODIS images, (2) the need for temporal and spatial coincidence with SWOT tracks, and (3) the requirement that ISWs propagate in regions free of mesoscale eddy influence. In particular, sunglint images are especially limited by cloud cover and suitable solar illumination conditions required to detect ISWs, which significantly reduce the number of usable scenes in this region
* * *
**Comment 5:** « There are two "section 5.5" in this paper, with one appearing ahead of "section 5.4". »

**Response 5:** Thank you for this comment.We have carefully revised the manuscript to correct the section numbering, ensuring that the sequence now follows the proper order without duplication.

---

## Author Comment (AC2)

**Subject: Response to Anonymous Referee #2 comments          9 Dec 2025**

Dear Madam/Sir,

We sincerely thank you for taking the time to review our manuscript, *"Internal Solitary Waves Refraction and Diffraction from Interaction with Eddies off the Amazon Shelf from SWOT"* and to provide your comments and suggestions for enhancing the quality of this paper. Below, we present a detailed, point-by-point response.

**Response to Anonymous Referee #2**

**Comment 1 :**

**a)** « One concern that I have had is how the background conditions are interpreted. The NE and CE case appear to represent the same mesoscale background field (NE: 18 Sep 2023; CE: 29 Sep 2023). In other words, the cyclone and anticyclone detected in the CE case are also evident in the NE case. Just because the eddy tracking does not identify those as eddies, the ADT and associated currents highly in Figure 5a and 5b suggest eddy or mesoscale activity, don't they?

**Response a )** We thank the reviewer for this pertinent observation and share their concern regarding the interpretation of the background conditions. In our study, the NE case is defined as a "no-eddy" case because the Py-Eddy-Tracking algorithm did not identify any closed contour fulfilling all required criteria (minimum number of pixels, amplitude, shape, and number of extrema). Although weak mesoscale signatures are visible in the NE region, they do not correspond to fully developed eddies. We have extensively tested the sensitivity to this issue and have selected a new showcase presented below

**b)** Are the associated currents in the NE case significantly weaker than in the CE case?

**Response b )** At the reviewer's request, we computed the geostrophic currents shown in **Figure d**. The analysis of the velocity field indicates that the currents are weaker and exhibit fewer meanders in the NE case than in the CE case, which is consistent with the absence of fully developed eddies on 18 September 2023.

[Figure]

*Figure d. Geostrophic velocities in the NE case and CE case.*

**c)** I guess that for the given day (18 Sep 2023) there are no enclosed ADT contours for the anticyclone and cyclone in ISW propagation direction? The eddy detection considers only local extrema of ADT?

**Response c )** To clarify how the Py-Eddy-Tracking algorithm operates: it identifies local extrema of the ADT field and all closed contours that enclose them. **Figure e** shows the detected contours in the NE case, along with the associated rejection criteria. Although closed ADT contours are indeed present at 43° W–5° N over the seamount, they were discarded because they do not satisfy the amplitude criterion and contain two extrema within a single closed contour.

[Figure]

**Figure e.** All Contours detected in the NE case. Green contours represent accepted eddies, while red, yellow, blue, and black contours correspond to rejections due to shape error, amplitude criterion, masked values within the contour, and pixel size limits, respectively.

Finally, based on these verifications, we selected another showcase to illustrate the "no-eddy interaction" (NE) case. As already described in our response to Reviewer #1, we selected SWOT pass 227, cycle 012, which sampled the ISW fluxes generated at site A, offshore of the Amazon, on 14 March 2024 (**Figure f**).These figure is now described section 'Signature of ISW, refraction and diffraction from the interaction with eddies' (section 4.2) lines 296 -330.

The spectral analysis of the ISW fluxes sampled before the seamount interaction area (**Fig. g.A, black spectrum #1**) shows two peaks at wavelengths of 180–140 km and 75 km, corresponding to mode-1 and mode-2 IT wavelengths, respectively. After the ISWs cross the seamount, the spectrum displays enhanced energy at scales below 50 km, with peaks between 30–40 km, suggesting the presence of mode-3 IT (**Fig. g.A, red spectrum #2**).

We then extended the analysis to characterize the wave trains. In the NE case, the wavelength spectra show no significant peak in area 1', indicating the absence of secondary structures near the principal wave crest within the ISW packet (**Fig. g.D, black spectrum #1'**). However, the spectra around the subsequent individual ISW packets from the A–D flux are dominated by components at 20 km (**Fig. g.D, black spectrum #1"**). Spectral analysis figures is now described section 'Spectral analysis : dominant wavelength' (section 4.3.2) lines 353 -368.

After reconstructing the wave crests, no change in trajectory or curvature is observed before or after the seamount. More description are develop in text in following section : 'Crest to crest distance variability and ISWs-mode shifts' ( section 4.3.3) line 381-383 and line 388-390 ; 'Wavecrest geometry and direction of propagation' ( section 4.3.4 ) line 399- 404

In conclusion, this exploration suggests that the response of the baroclinic IT flux depends on the amplitude of the eddy. Indeed, in both cases classified as NE (the previous and the newly selected one), where the geostrophic currents are weaker — amplitude below 0.046 m in the new NE case and below 0.034 m in the old NE case — the IT flux does not appear to be modified. In contrast, the CE and AE cases, where amplitudes exceed 0.066 m in CE and 0.056 m in AE, exhibit a clearly distinct behavior.

[Figure]

**Figure f**: *Eddy detection maps based on MIOST L4 ADT and ADT_swot SWOT cycle 012, passes 227, from 2024-03-14 combined with Level 1B optical imagery MODIS-Aqua from 2024-03-10. Cyclonic eddies and anticyclonic eddies are marked by red and green circles, respectively.*

[Figure]

*NE=no eddy. Bathymetry is represented using isocontours at –400 m, –3000 m, –100 m, and 0 m*

***Figure g:*** *Mean power spectrum density of SWOT ADT_swot along track for each area for G) NE 2024/03/14. Black (red) lines refer to spectrum located before (after) interaction with seamount. Dotted line refers to spectrum of single wave packet and solid line spectrum of ISWs . Area number is indicated between parenthesis. ADT_swot with ISWs detection NE 2024/03/14. NE=no interaction eddy case. Bathymetry is represented using isocontours at –3500 m, –3000 m, –100 m, and 0 m*

**d)** Is it possible to add the velocity field (local normalized angular momentum) as it is done in the AMEDA algorithm (Le Vu et al., 2018). I assume that the authors have not found a better example for NE with corresponding SWOT passes. The authors may stick with the given example but change the definition of NE, which for now is defined as "absence of mesoscale eddies" to something like "absence of interaction...". »

**Response d )** The inclusion of the velocity field (normalized angular momentum), as used in the AMEDA algorithm, is not implemented in the Py-Eddy-Tracker code. While it is technically possible to add this variable, doing so would require substantial development work. An important point — which strengthens the robustness of our results — is that we chose to rely on the detection of closed sea-level anomaly contours rather than on a velocity-based metric, because geostrophic velocities become less reliable as one approaches the equator.

Following the reviewer's recommendation, we also decided to rename our NE (No Eddy Interaction), CE (Cyclonic Eddy Interaction), and AE (Anticyclonic Eddy Interaction) cases to emphasize the interaction, which is at the core of our scientific question.
* * *
**Comment 2 :** « Following up the above a comment, there is a lot of speculation on the ISW propagation direction after encountering seamounts and eddies, though this is well discussed and

limitations of the presented manuscript are clearly listed (see Discussion). A ray tracing experiment that models the horizontal propagation of tidal rays (separated by vertical modes) through a mesoscale eddy field (see Rainville and Pinkel, 2006; Vic et al., 2023; Bendinger et al., 2024 or a more dedicated approach by Duda et al., 2018, Guo et al., 2023) would certainly remove any doubt on the ISW propagation direction proofing whether the observed mesoscale eddies (CE and AE) explain indeed the observed ISW refraction in propagation direction and whether NE causes no refraction in propagation direction. Such ray tracing could be applied to the three surface geostrophic velocity fields in Figure 7D to 7F (or Figure 12) to see whether it matches the changing propagation of ISWs observed in SWOT and sunglint images. However, I understand that this involves further work, which is why I leave it to the authors whether they want to quantify their findings with theory. »

**Response 2 :** We appreciate the reviewer's insightful comment and the valuable suggestion regarding ray-tracing experiments. Performing ray tracing that includes background currents would indeed help disentangle the respective roles of topography and currents in the diffraction and scattering responses we observed. However, implementing such an analysis represents a substantial amount of additional work and would warrant a dedicated study, which we may consider in the future.

To our knowledge, several complementary studies also address this issue. Among them, two have already been published: Dunphy et al. (2014) and Wang & Legg (2023), which show that ISW trajectories are modulated by eddies and support our results. In addition, two other studies from our group are currently in preparation and will specifically address this question.

- Kougang et al. 2025b used realistic NEMO simulations in the AMAZON-39 configuration, to investigated the interactions between internal tides and mesoscale eddies in the same region. These results are part of a separate study that will be submitted before the end of the year (Kuogang et al. 2025b).

- Kougang et al. 2026 investigated in the region using idealized CROCO simulations pour etudier the occurrence of refraction and diffraction induced by vortices, with responses modulated by vortex characteristics such as radius, vertical extent, azimuthal velocity, and vorticity.

In the new manuscript, we added in the section conclusion/perspectives (section 6) at **l.593** : *« A comparison with results from idealized and ray-tracing experiment that simulates the horizontal propagation of internal tidal rays tthrough a mesoscale eddy field might highlight the IT propagation direction.*
* * *
**Comment 3:** « Before focusing on how eddies impact ISW characteristics, I would start with a small paragraph describing their mean characteristics (based on Table 1). How many ISW were detected, etc.? I think it is also worth to quantify their SSH signature (>10 cm) ? »

**Response 3 :** We completely agree we this comment and have added a specific paragraph. This actually constitutes now our first result, which is mentioned in the abstract, conclusion and have been detailed on new section 'Spectral analysis : ISWs mean characteritics ( section 4.3.1)' line 340-350. Note that a specific work has been done on the amplitude of the wave in Da Silva et al. 2025.
* * *
**lines 24-26:** Personally, I think that the characterization of ISWs using SWOT deserves a bit more attention in the abstract (and in the conclusion also), stating at least how many ISW you detected and what their SSH amplitude is.

**Response lines 24-26** : As for the previous comment we agree with this comments and have added new ISWs characeristic 'abstract' lines 18 to 46.
* * *
**line 41:** in a stratified ocean

**Response l.41** : Correction has been made *l.41 « in a stratified ocean. »*
* * *
**line 62:** and throughout the manuscript: To be double checked with the Copernicus editorial service whether work not submitted/published should be mentioned as reference. If so, it would be great to show associated results of those studies in preparation wherever they are relevant in the given manuscript. This could be in form of appendix, etc.

**Response line 62** : We appreciate the reviewer's interest in the comparison with the results of Kuogang et al., 2025b. However, these results are part of a separate study that is still in preparation and have not yet been publicly released. Therefore, we are unable to share the corresponding figures or datasets at this stage. We will also include a reference to the forthcoming paper once it becomes publicly available.
* * *
**line 77 and 81:** The authors introduce the terms coherent and incoherent. Briefly define what coherent and incoherent means for readers which are not familiar.

**Response line 77 and 81:** Correction has been made succinctly in the former text *l.79-85 « Consequently, the internal tide flux remains relatively stable and coherent **(the phase-locked, relatively stable part of the IT**) . From August to December (ASOND), the pycnocline deepens, the river discharge decreases, and NBC intensifies (Silva et al., 2005; Aguedjou et al., 2019; Tchilibou et al., 2022), which forms NECC. Instabilities in these currents generate a series of cyclonic and anticyclonic eddies (Garzoli et al., 2004). These structures significantly modify ISW propagation, trajectory, speed, amplitude, geometry,  interpacket distance, and increase the incoherent, component of the internal tide **(the non-phase-locked, time-varying part of IT**, Bendinger et al, 2025 ; Dunphy and Lamb, 2014; Ponte and Klein, 2015; Dunphy et al., 2017; Wang and Legg, 2023; Xie et al. ,2015; Xu et al., 2020; Huang et al., 2024). »*
* * *
**Figure 1 and 2:** Figure 1 and 2 could be merged to one figure with two subplots?

**Response Figure 1 and 2:** Correction has been made *l.106*

[Figure]

*« Figure 1 : A) Bathymetry of Amazon shelf from 0 to –5000 m. IT generation sites labeled A to F along the shelf break. Black solid contours delineate a typical area where ISWs propagation is observed from sites A and D. The NBC and NECC are highlighted with thick grey arrows. Cyclonic eddies (CE) and anticyclonic eddies (AE) are marked respectively by red and green circles. Seamounts are delineated by 4000 m and 3300 m isobaths. B) The color map shows the 25-hour mean depth-integrated baroclinic internal tide energy flux from the NEMO model from September 2015 (Assene et al., 2024), radiating from IT generation sites labeled A to F along the shelf break. ISW surface signatures (black dotted lines) detected in MODIS/TERRA satellite imagery from De Macedo et al. (2023). Black solid contours delineate a typical area where ISWs propagation is observed from sites A and D. »*
* * *
**line 129:**

**a)** MIOST maps include SWOT KaRIn and nadir observations. Do the authors expect that submesoscale and wave-like motion is contained in ADT (when SWOT swaths are available and used for the multiscale, multivariate mapping), which was used to apply the mesoscale eddy tracking? In other words, when studying mesoscale dynamics should one not exclude the SWOT KaRIN observations?

**Response a) :** This is a relevant question that we asked ourselves. The MIOST mapping method might separates submesoscale processes present in the KaRIN data and should retain only the geostrophic component. Therefore, we do not expect to observe submesoscale or wave-like motions in the MIOST maps.

One disadvantage of the MIOST map is that, if this separation is not done properly, the MIOST map (including SWOT) may contain noise introduced by ISWs, which could be misinterpreted as fronts or filaments.

At present, an advantage of the MIOST map (including SWOT) is that it provides higher effective resolution and a more accurate representation of mesoscale eddies (their location and intensity) than DUACS products, thanks to its mapping method and the integration of the SWOT swaths (for example, Ballarotta et al., 2024).

In conclusion, the MIOST products with and without the SWOT swath may differ and may be affected by ISWs. To assess this sensitivity, we tested our results on the MIOST map without SWOT and compared them to the MIOST map with SWOT.

**b)** I think that there is a dataset provided by AVISO/CMEMS which provides ADT maps but without SWOT. I do not expect big differences for the final eddy tracking, but this contrasts with what was being said in lines 91-94, i.e. SWOT should be used with caution when studying mesoscale and geostrophic currents.

**Response b) :** AVISO/CMEMS without SWOT is not available for public access, but I was fortunate to have access to the MIOST product without SWOT ( only for 2023 year ) thanks to sensitivity tests that have been conducted in the CLS group (pers. comm. Ballarota). This product includes only nadir altimeters from CryoSat-2, HY-2, Jason-3, Sentinel-3A, Sentinel-3B, and Sentinel-6A and is shown **Figure h.2** for the CE case and is compared to the product including SWOT (**Figure h.1**). In this case, the difference in ADT between the two products is shown **Figure h.3**. We mainly see that the general pattern (number of eddy detected, radius, intensity, etc) and eddy structure remain unchanged, even if locally some high differences of SSH can be found.

[Figure]

*Figure h :* *Eddy detection on MIOST map including SWOT KaRIN (1) and excluding SWOT KaRIN (2). Residual ADT map (3) from difference between (1) and (2).*

**c)** However, in line 138 it is implied that ADT maps consist of nadir-pointing data only. Please clarify how SWOT data (KaRIn and nadir) is implemented in ADT maps.

**Response c)** Thanks for pointing out this incoherency in our paper. We have corrected this inconsistent sentence at line *l.138 « The spatial resolution of MIOST maps is too low to resolve submesoscale processes, such as ISWs".* As explained previously, the MIOST maps include both nadir altimeters and SWOT KaRin mesurment.
* * *
**Figure 2:** It seems like, transparency has been added to the colorbar, but the shading in the actual plot has no transparency ? It could be a nice addition to plot all ISWs identified in this manuscript on top of those from Macedo et al. (2023) ?

**Response Figure 2 :** After verification, there is no transparency effect on the colorbar compared to the plot. For better visualization, we have remade the figure using more opaque colors. We also add all ISWs identified in this study on top of those from Macedo et al. (2023) as you sugest.

[Figure]

**Response line 145:** Correction has been made *l.145 « 120 km acros the track »*
* * *
**Section 2.2**:

**a)** I assume that the authors used the 250 m (unfiltered) product? Please clarify. If yes, also mention in the introduction in line 87 that observations are at disposal down to 250 m.

**Response a :** This is an important point that we have clarified**.** We did NOT use the 250 m product, but the SWOT Level-3 LR Expert v2.0.1 product, which has a sampling spatial resolution of 2 km. For clarity, we will add the following sentence to the manuscript: *l.152 « on a 2 km spatial grid spacing. »*

**b)** Did the authors compare the 250 m and 2 km products? If yes, are there big differences between these two?

**Response b :**We estimated that the 2 km resolution was sufficient to meet the objectives of this study. Moreover, the production of the 250 m product was delayed compared to the 2 km product, and at the time of our analyses, the 250 m product did not yet cover the 2024 period. Therefore, we did not compare the two products in our show cases.

**c)**I expect that the ISWs might have a stronger (unsmoothed) signature in the 250 m product?

**Response c :**In fact, we expect that smaller-scale wave trains would be observable in the 250 m product, as shown, for example, in Cheshme et al. (2025).
* * *
**line 158:** Specifically state what is included in the high-resolution signal, e.g. tidal and non-tidal IGWs, submesoscale, etc.

**Response line 158:** Correction has been made *l.158 « Therefore, ADT_swot contains all the high-resolution signal not resolved by the corrections applied to the KaRIn data,as well as the part of the signal not resolved by the MIOST mapping method, including sub-mésoscale eddies, fronts, filaments, ISWs, internal gravity waves, wind waves, etc»*
* * *
**line 181-183:**

**a)** Is this commonly done when applying eddy tracking in near-equatorial regions?

**Response a :** In near-equatorial regions, eddy tracking can be challenging due to the small value of the Coriolis parameter, which limits the validity of the geostrophic approximation. For this reason, we use a detection algorithm based on the identification of closed sea surface height contours around ADT extrema. Similar algorithms, such as Py-Eddy-Tracker, have been applied in other studies near the equator; for example, Aguedjou et al. (2019) and Dossa et al. (2022) performed eddy tracking in the tropical Atlantic Ocean.

**b)** Why not using SLA maps when focusing on mesoscale eddies?

**Response b :** Moreover, we perform the detection on ADT rather than SLA because Peliasco et al. (2021) recommend using ADT fields for mesoscale structure detection, as they better capture these features compared to SLA. In the new manuscript we have added in section 'Eddy Detection Method' ( section 3.1) the following text : *l.188 "Mesoscale eddies were detected from ADT fields, as recommended by Peliasco et al, 2021"*
* * *
**line 202:** Fig. 3A?

**Response line 202 :** Correction has been made *l.202 « ( Fig. 3.A., black spectrum ) »*
* * *
**Figure 3:** I would enlarge the figure/subplots for the better visibility.

**Response Figure 3 :** We are agree with your comment. The figure was enlarged by arranging the subplots vertically.
* * *
**Figure 6:** I leave it to the authors whether they want to keep this figure. For me the text is sufficient.

**Response Figure 6 :** Thank you for this comment, but we have chosen to keep this figure. We have adapted it to the new NE case (14 March 2024).

[Figure]

**Section 4.3:** ISW propagation is interpreted using surface geostrophic velocities, mainly representative of mode-1. What role could higher-baroclinic current velocities play (see Duda et al., 2020, Guo et al., 2023)?

**Response section 4.3.** This is a very interesting question. Indeed, it is expected that baroclinic currents can redirect, and even trap, internal tides, inducing energy transfer, as shown by Duda et al. (2020) and Guo et al. (2023). With our satellite altimetry surface data, we have only access to barotropic component or the baroclinic mode-1 structure, which represents a major limitation for detecting and characterizing baroclinic eddies. These observational limitations may lead to an underestimation of the extent and complexity of eddy-induced ISW modifications.

Moreover, we searched for ARGO data in our two cases :CE and AE to characterize the vertical structure of the eddies at the core of their interactions with the ISWs. Unfortunately, no float sampled the eddies in our showcase cases.

However, in our group, several upcoming studies are currently addressing this issue. Indeed, in this region baroclinic currents and vertical dipoles are suspected (Carton et al., 2021). The recent AMAZOMIX moorings that have been recovered, under analysis, might enlighten this question.

We have added this point in the Discussion section 'Satellite Sampling' (Section 5.3.2) : **line 545-554**:
« Finally, satellite altimetry and surface geostrophic velocities primarily capture the barotropic signal and the baroclinic mode-1 structure, which remains a substantial limitation for the detection and characterization of baroclinic eddies. Yet, the vertical structure and intensity of baroclinic eddies likely condition their interactions with the surrounding flow and with propagating internal solitary waves, potentially leading to distinct dynamical responses. Our results provide qualitative and first-order evidence of eddy–ISW interactions as observed by SWOT, but the absence of in situ measurements prevents us from assessing the vertical structure of the eddies involved. It is therefore reasonable to expect that these interactions may differ when eddies exhibit a more complex vertical configuration, such as a dipolar structure, and future work should aim to resolve and analyze this vertical structure to better quantify these processes.

These observational limitations may lead to an underestimation of the extent and complexity of eddy-induced ISW modifications and emphasize the need to complement the analysis with in situ measurements or 3D high-resolution modeling to better capture the full scale and dynamics of ISWs and their interactions with eddies »
* * *
**Section 4.3.1:** ISWs are highly anisotropic. The significance/meaningfulness of (along-track) wavenumber spectra depend on whether the selected tracks are aligned with the primary propagation direction. The latter seems mostly to be the case. Whatsoever, I believe that this is worth mentioning. SWOT cycle 20, for the most easterly pass it could be possible that the tracks used for the spectral analysis are partly aligned with the wavecrests?

**Response Section 4.3.1:** Thank you for this very relevant comment. Indeed, ISWs are highly anisotropic, and the along-track wavenumber spectrum necessarily depends on the angle between the SWOT sampling direction ($\theta_{SWOT}$) and the actual propagation direction of the ISW packets ($\theta_{ISW}$). In most cases, the selected tracks are indeed close to the main propagation axis ($\theta_{ISW/SWOT} < 30°$). However, this is not the case for track 20 (AE case). In this case, $\theta_{ISW/SWOT} > 60°$. Consequently, $\cos(\theta_{diff}) < 0.50$, which results in a difference of $D_{SWOT} = 40$ km, $D_{ISW}=20$ km, following : $D_{true} = D_{obs} \cdot \cos(\theta_{diff})$ (Results for each case are details in **Table a).** Consequently, in the previous version we are in the referentiel of SWOT nadir axe, whereas now thanks to your comment to also depict the wavelength in the referential of the ISW propagation axe. We have modified the text to distinguish between $D_{SWOT}$ and $D_{ISW}$ in the Materials & Methods (line 249-259), and detailed $D_{ISW}$ in Results sections (ex line 391 line 394, etc), Table 1 (line 343) and synthetic schematic (Figure 12, line 619).

Table a : Characteristics of ISWs detected during (NE) 2024/03/13, (CE) 2023/09/29 and (AE) 2024/08/22

| | NE | | | CE | | | | AE | | | | | |
|---|---|---|---|---|---|---|---|---|---|---|---|---|---|
| SWOT | 012/227 | | | 004/227 | | | | 020/074 | | | 020/046 | | 020/018 |
| Area (°N) | 1 | 1'' | 2 | 3 | 3' | 3'' | 4 | 5 | 5' | 6 | 8 | 8' | 9' |
| $\theta_{swot}$ | 8.55 | 8.55 | 8.55 | 8.66 | 8.66 | 8.66 | 8.66 | -8.56 | -8.56 | -8.56 | -8.53 | -8.53 | -8.53 |
| $\theta_{ISW}$ | 24.3 | 27.2 | 18.5 | 29.7 | 29.7 | 29.7 | -20.4 | 25.6 | 39.8 ; 35 | -1.6 | 8.57 | 51.3 | 52.3 |
| Cos ($\theta_{ISW/swot}$) | 0.96 | 0.94 | 0.98 | 0.93 | 0.93 | 0.93 | 0.87 | 0.82 | 0.66;0.72 | 0.99 | 0.95 | 0.50 | 0.48 |

| | | | | | | | | | | | | | |
|---|---|---|---|---|---|---|---|---|---|---|---|---|---|
| **D** $\mathbf{D_{swot}}$ | 140 | 20 | 30 | 140 | 12 | 15-20 | 40-35 | 154 | 20-25 | 10-12 | 10-12 | 40-25 | 40-25 |
| $\mathbf{D_{ISW}}$ | 134.4 | 18.8 | 29.4 | 130 | 11.1 | 14-18 | 30-35 | 126 | 18-13 | 10-12 | 9.5-11 | 20-12.5 | 19-12 |
* * *
**Caption Figure 8:** Subplots D-F are not mentioned.

**Response Caption Figure 8 :** Thank you for this comment. Correction had be made.
* * *
**Section 5.2.3 :** Important section discussing the NECC. Why is there a big interest in separating effects of NECC and eddies? Does it matter whether the NECC or eddies refract/diffract ISWs? A ray tracing experiment could possibly separate these two effects when modeling the propagation of tidal rays in climatological/annual mean background current 2 fields (which represent the NECC?) and daily ADT/SLA maps (which represent eddies).

**Response Section 5.2.3 :** Thank you for your comment. The eddies present in this region are largely generated by the meanders of the NECC, so the two phenomena are closely linked. In this context, strictly distinguishing between "the effect of the NECC" and "the effect of the eddies" does not necessarily make much sense if our ultimate goal is simply to understand how the total current field refracts or diffracts the ISWs. One reason to distinguish the effects of the NECC and the eddies lies in the possibility of predicting their respective impacts, given that their variability, vertical extent, and amplitude can differ or vary significantly. Several idealized modeling studies focusing on interaction processes simulate either the effect of a current (Duda et al., 2020; Guo et al., 2023) or the effect of an eddy (Wang and Legg, 2023). Future studies evaluating ray tracing or using idealized models that include both an eddy and a meander could help better quantify the respective impacts of each.
* * *
**lines 446-450:** Following a comment from above, it would be very helpful to show something from Kouogang et al. (2025b, 2025c in preparation).

**Response lines 446-450:** These results are part of a separate study that is still in preparation and have not yet been publicly released. Therefore, we are unable to share the corresponding figures or datasets at this stage. We will also include a reference to the forthcoming paper once it becomes publicly available.
* * *
**lines 450:** Duda et al. (2018) is a modeling study, isn't it? In that case prevent using "observed".

**Response line 450 :** Correction has been made *l.480 « For comparison, it has been showed that the meandering Gulf Stream significantly refracts and traps ITs (Duda et al., 2018) »*
* * *
**lines 552-553**: Not as detailed as in the presented study, but I think it is worth mentioning the studies Xie et al. (2015), Xu et al. (2020), and Huang et al. (2024) who simultaneously observed ISWs and mesoscale eddies. These references might be also added to the introduction.

**Response lines 552-553 :**Tkank you for your suggestion. We had these citations in the introduction. *l.85-98* *"These structures significantly modify ISW propagation, trajectory, speed, amplitude, geometry, interpacket distance, and increase the incoherent, component of the internal tide (the non-phase- locked, time-varying part of IT ; Bendinger et al, 2025 ; Dunphy and Lamb, 2014; Ponte and Klein, 2015; Dunphy et al., 2017; Wang and Legg, 2023, Xie et al. (2015), Xu et al. (2020), Huang et al. (2024))."*
* * *
**lines 630-632:** There is now a peer-reviewed version: https://doi.org/10.5194/os-21-1943-2025

**Response line 630-632:** Correction has been made *l.674-676* *« Bendinger, A., Cravatte, S., Gourdeau, L., Vic, C., and Lyard, F.: Regional modeling of internal-tide dynamics around New Caledonia. Part 2: Tidal incoherence and implications for sea surface height observability , Ocean Sci., 21, 1943–1966, https://doi.org/10.5194/os-21-1943-2025, 2025 »*
* * *
**lines 650-653:** Remove the doi of the preprint.

**Response line 650-653:** Correction has been made *l.694-697* *« De Macedo, C. R., Koch-Larrouy, A., Da Silva, J. C. B., Magalhães, J. M., Lentini, C. A. D., Tran, T. K., Rosa, M. C. B., and Vantrepotte, V.: Spatial and temporal variability of mode-1 and mode-2 internal solitary waves from MODIS/TERRA sunglint off the Amazon shelf, Ocean Sci, 19, 1357–1374, https://doi.org/10.5194/egusphere-2022-1482, 2023 »*
* * *
**lines: 715-719:** To be double checked with the Copernicus editorial service whether work which is not submitted/published should be in the reference list

**Response line 715-719:** We will also include a reference to the forthcoming paper once it becomes publicly available.